# UTC-IE: A Unified Token-pair Classification Architecture for Information Extraction

## Abstract

Information Extraction (IE) spans several tasks with different output structures, such as named entity recognition, relation extraction and event extraction. Previously, those tasks were solved with different models because of diverse task output structures. Through re-examining IE tasks, we find that all of them can be interpreted as extracting spans and span relations. We propose using the start and end token of a span to pinpoint the span in texts, and using the start-to-start and end-to-end token pairs of two spans to determine the relation. Hence, we can unify all IE tasks under the same token-pair classification formulation. Based on the reformulation, we propose a **U**nified **T**oken-pair **C**lassification architecture for **I**nformation **E**xtraction (**UTC-IE**), where we introduce Plusformer on top of the token-pair feature matrix. Specifically, it models axis-aware interaction with plus-shaped self-attention and local interaction with Convolutional Neural Network over token pairs. Experiments show that our approach outperforms task-specific and unified models on all tasks in 10 datasets, and achieves better or comparable results on 2 joint IE datasets. Moreover, UTC-IE speeds up over state-of-the-art models on IE tasks significantly in most datasets, which verifies the effectiveness of our architecture.

## 1 Introduction

Information Extraction (IE) aims to identify and classify structured information from unstructured texts (Andersen et al., 1992; Grishman, 2019). IE consists of a wide range of tasks, such as named entity recognition (NER), joint entity relation extraction (RE)[1] and event extraction (EE) [2].

In the last decade, many paradigms have been proposed to solve IE tasks, such as sequence labeling (McCallum & Li, 2003; Huang et al., 2015; Zheng et al., 2017; Yu et al., 2020a), span-based classification (Jiang et al., 2020; Yu et al., 2020b; Wang et al., 2021; Ye et al., 2022), MRC-based methods (Levy et al., 2017; Li et al., 2020; Liu et al., 2020) and generation-based methods (Zeng et al., 2018; Yan et al., 2021a; Hsu et al., 2022). The above work mainly concentrates on solving individual tasks, but it is desired to have a unified model to solve all IE tasks without designing dedicated modules. Besides, tackling all IE tasks with one model can facilitate knowledge sharing between different tasks. Therefore, various attempts have been made to unify all IE tasks with one model structure. Wadden et al. (2019); Lin et al. (2020); Nguyen et al. (2021) encode all IE tasks' target structure as graphs and design graph-based methods to predict them; Paolini et al. (2021); Lu et al. (2022) solve general IE tasks in a generative way with a text-to-text or text-to-structure framework. However, graph-based models tend to be complex to design, and generative models are time-consuming to decode.

In our work, we creatively propose a simple yet effective paradigm for unified IE. Inspired by Jiang et al. (2020), we re-examine IE tasks and consider that all of them are fundamentally *span extraction* (entity extraction in NER and RE, trigger classification and argument span detection in EE)

---

[1] Joint entity relation extraction aims to extract both entities and relations. In our paper, we call it relation extraction (RE) for simplicity.

[2] Event extraction covers trigger extraction and argument extraction, where we first conduct argument span detection and then conduct argument role classification in our architecture.

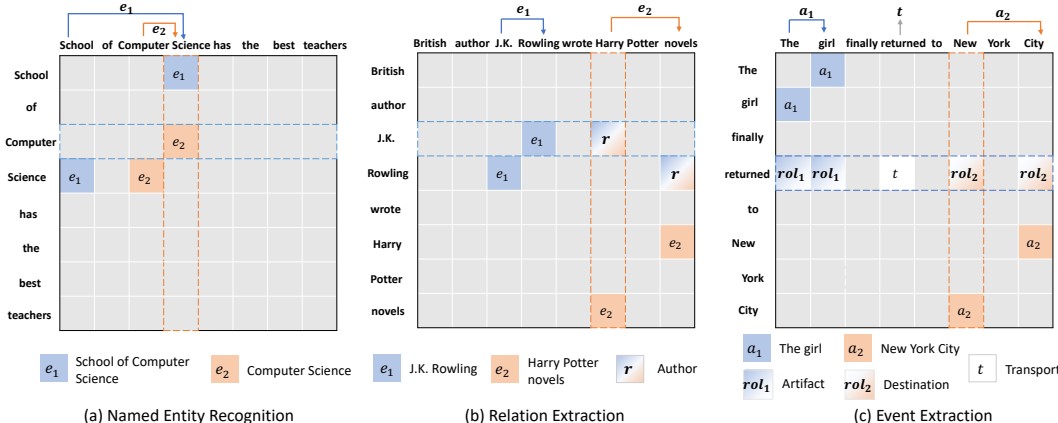

Figure 1: An illustration of the token-pair decomposition for IE tasks. Each cell represents one token pair, and it can be classified into pre-defined types. $e$, $r$, $t$, $a$ and $rol$ in figures mean entity, relation, event trigger, event argument and event role. For the span extraction, we use the start-to-end and end-to-start token pairs to pinpoint the span, such as entity spans $e_1, e_2$, argument spans $a_1, a_2$ and trigger span $t$ (cells with pure color). For the relational extraction, we use the start-to-start and end-to-end token pairs to represent the relation, such as $r$ and $rol_1, rol_2$ (cells with gradient color). Therefore, all IE tasks can be decomposed into token pair classifications. After the reformulation, the local dependency and interaction from the plus-shaped orientation (as the orange and blue dotted lines depict) can provide vital information to classify the central token pair.

or *relational extraction*[3] (relation extraction in RE and argument role classification in EE). Based on this perspective, we further simplify and unify all IE tasks into *token-pair classification tasks*. Figure 1 shows how each task can be converted. Specifically, a span is decomposed into start-to-end and end-to-start token pairs. As depicted, the entity "School of Computer Science" in Figure 1(a) is decomposed into indices of (School, Science) and (Science, School). As for detecting the relation between two spans, we convert it into start-to-start and end-to-end token pairs from head mention to tail mention. For example, in Figure 1(b), the relation "Author" between "J.K. Rowling" and "Harry Potter novels" is decomposed into indices of (J.K., Harry) and (Rowling, novels).

Based on the above decomposition, we propose a **U**nified **T**oken-pair **C**lassification architecture for **I**nformation **E**xtraction (**UTC-IE**). Specifically, we first apply Biaffine model on top of the pre-trained language model to get representations of token pairs. Then we design a novel Transformer to obtain interactions between them. As the plus-shaped dotted lines depicted in Figure 1, token pairs in horizontal and vertical directions cover vital information for the classification on the central token pair. For span extraction, token pairs in the plus-shaped orientation are either clashing or nested with the central token pair, for example, $e_2$ is contained by $e_1$ in Figure 1(a); for relational extraction, the central token pair's two constituent spans locate in the plus-shaped orientation, such as in Figure 1(b), $r$ is determined by $e_1$ and $e_2$. Therefore, we make one token pair only attend horizontally and vertically in the token pair feature matrix. In addition, position embeddings are incorporated to keep the token pairs position-aware. Moreover, neighboring token pairs are highly likely to be informative to determine the types of the central token pair, so we apply Convolutional Neural Network (CNN) to model the local interaction after the plus-shaped attention. Since the attention map for one token pair is intuitively similar to the plus operator, we name the novel module as **Plusformer**.

We conduct numerous experiments in two settings. When training separately on each task, our model outperforms previous task-specific and unified models on 10 datasets of all IE tasks. When training a single model simultaneously on all IE tasks in one dataset (named as joint IE task), UTC-IE achieves better or comparable results than 2 joint IE baselines. To thoroughly analyze why our UTC-IE architecture is useful in IE tasks under the token-pair paradigm, we execute several ablation studies. We observe that CNN module in Plusformer plays a significant role in IE tasks because of the abundant local dependency between token pairs after the reformulation. Furthermore, owing to

---

[3]In this paper, we use relational extraction to represent extracting relations between spans, which has broader meanings than relation extraction.

the good parallelism of self-attention and CNN, UTC-IE is one to two orders of magnitude faster than prior unified IE models and some task-specific work.

To summarize, our key contributions are as follows

1. We introduce UTC-IE, which decomposes all IE tasks into *token-pair classification tasks*. In this way, we can unify all IE tasks under the same task formulation. Henceforth, we can use one model to fit all IE tasks without designing task-specific modules. Besides, this unified decomposition is much faster than recently proposed generation-based unified frameworks.

2. After the reformulation of different IE tasks, we propose the Plusformer to model interaction between different token pairs. The plus-shaped self-attention and CNN in Plusformer are well-motivated and effective in the reformulated IE scenario. Experiments in 12 IE datasets all achieve state-of-the-art (SOTA) performance which justifies the superiority of Plusformer in IE tasks.

3. The reformulation enables us to use one model to fit all IE tasks. Therefore, we can train one model on three IE tasks, and results on two joint IE datasets show that the proposed unification can effectively benefit each IE task through multi-task learning.

4. Extensive ablation experiments reveal that components in Plusformer are necessary and beneficial. Among them, CNN module in Plusformer can be essential to the overall performance. Analysis shows that this performance gain is well-explained because when reformulating IE tasks into token pair classifications, the adjacent token pairs can be informative and CNN can take good advantage of the local dependency between them.

## 2 TASK DECOMPOSITION AND DECODING

We first introduce how we decompose IE tasks to conduct training, and then present the decoding procedure for the decomposition. More discussions about the decomposition are presented in Appendix A.

### 2.1 TASK DECOMPOSITION

Formally, given an input sentence of $L$ tokens $\boldsymbol{x} = [x_1, x_2, ..., x_L]$, the potential token pairs can form a score matrix $\boldsymbol{Y} \in \mathbb{R}^{L \times L \times (|\mathcal{S}| + |\mathcal{R}|)}$, where $\mathcal{S}$ is span classes, $\mathcal{R}$ is relational classes. We stipulate

- When a span $(s, e)$ is of type $t$, then $\boldsymbol{Y}[s, e, t] = \boldsymbol{Y}[e, s, t] = 1$, where $s, e \in [1, L]$ and $t \in [1, |\mathcal{S}|]$ are the start, end token indices and span type;

- When the span $(s_1, e_1)$ forms the relation $r \in [|\mathcal{S}| + 1, |\mathcal{S}| + |\mathcal{R}|]$ with another span $(s_2, e_2)$, then $\boldsymbol{Y}[s_1, s_2, r] = \boldsymbol{Y}[e_1, e_2, r] = 1$.

NER aims to extract all entities $\{(s_i, e_i, t_i)\}$, where $t_i \in \mathcal{S}_e$ and $\mathcal{S}_e$ is pre-defined entity types. Therefore, in NER, $\mathcal{S} = \mathcal{S}_e$ and $\mathcal{R} = \phi$.

RE aims to extract all relations $\{((s_i^h, e_i^h, t_i^h), r_i, (s_i^t, e_i^t, t_i^t))\}$, where the superscript $h$ and $t$ denotes the head and tail entities, $t_i^h, t_i^t \in \mathcal{S}_e, r_i \in \mathcal{R}_r$ and $\mathcal{S}_e, \mathcal{R}_r$ are pre-defined entity types, and relation types. Therefore, in RE, $\mathcal{S} = \mathcal{S}_e$ and $\mathcal{R} = \mathcal{R}_r$.

EE aims to extract all events $\{\{(s_i, e_i, t_i), (s_{ia}^1, e_{ia}^1, rol_i^1), \ldots, (s_{ia}^k, e_{ia}^k, rol_i^k)\}\}$, where $(s_i, e_i)$ means the trigger span, $t_i \in \mathcal{S}_t$ is the event type, $\mathcal{S}_t$ is pre-defined event types; $s_{ia}, e_{ia} \in [1, L]$ are the start and end token indices of an argument span, $k$ is the number of arguments of the trigger, to extract argument spans, we set the argument type as $\mathcal{S}_a$, and $|\mathcal{S}_a| = 1$; $rol_i \in \mathcal{R}_o$ is the role type of the argument and $\mathcal{R}_o$ is pre-defined role types. We can view role types between the trigger and the arguments as relations. Therefore, in EE, $\mathcal{S} = \mathcal{S}_t \cup \mathcal{S}_a$ and $\mathcal{R} = \mathcal{R}_o$.

Joint IE aims to jointly extract entities, relations, and events in the text. Extracting entities and relations are generally the same as those in NER and RE. When extracting events, there is no need to extract argument spans purposely because all argument candidates are entities. Therefore, in joint IE, $\mathcal{S} = \mathcal{S}_t \cup \mathcal{S}_e$ and $\mathcal{R} = \mathcal{R}_r \cup \mathcal{R}_o$.

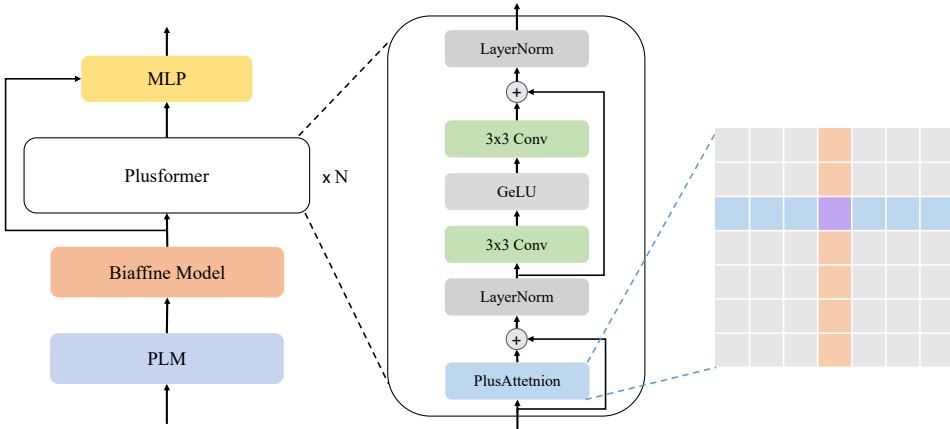

Figure 2: An overview of the UTC-IE Model.

## 2.2 DECODING

The decoding essentially extracts spans and relations from the score matrix $\boldsymbol{Y}$. If $\boldsymbol{Y}[s, e, t] = \boldsymbol{Y}[e, s, t] = 1$ and $t \in [1, |\mathcal{S}|]$, then the span $(s, e)$ is of type $t$. And for two spans $(s_1, e_1)$ and $(s_2, e_2)$, if $\boldsymbol{Y}[s_1, s_2, r] = \boldsymbol{Y}[e_1, e_2, r] = 1$ and $r \in [|\mathcal{S}| + 1, |\mathcal{S}| + |\mathcal{R}|]$, then the span $(s_1, e_1)$ forms relation $r$ with the span $(s_2, e_2)$. The above decoding is for the ideal situation, where no span clash exists. However, for model's predictions, we need to first resolve the conflicts. The decoding with model's predictions will be presented in Appendix B.

## 3 METHOD

Figure 2 shows an overview of the architecture. Firstly, we present Biaffine (Dozat & Manning, 2017) model based on pre-trained language models (PLM). Then, we propose a novel Transformer-like structure named Plusformer to model interactions between token pairs. Next, we describe loss functions.

### 3.1 BIAFFINE MODEL

Given an input sentence, we first apply a PLM as our sentence encoder to obtain the contextualized representation as follows

$$\boldsymbol{H} = [\boldsymbol{h}_1, \boldsymbol{h}_2, ..., \boldsymbol{h}_L] = \mathrm{PLM}([x_1, x_2, ..., x_L]), \tag{1}$$

where $\boldsymbol{H} \in \mathbb{R}^{L \times d}$, $d$ is the PLM's hidden size.

Next, we use the Biaffine mechanism to get features for each token pair as follows

$$
\begin{aligned}
\boldsymbol{H}^s, \ \boldsymbol{H}^e &= \mathrm{MLP}_{\mathrm{start}}(\boldsymbol{H}), \ \mathrm{MLP}_{\mathrm{end}}(\boldsymbol{H}), \\
\boldsymbol{S}[i, j] &= (\boldsymbol{H}^s[i])^T \boldsymbol{W}_1 \boldsymbol{H}^e[j] + \boldsymbol{W}_2(\boldsymbol{H}^s[i] \oplus \boldsymbol{H}^e[j]) + \boldsymbol{b},
\end{aligned}
\tag{2}
$$

where $\mathrm{MLP}_{\mathrm{start}}, \mathrm{MLP}_{\mathrm{end}}$ are multi-layer perceptron layers, $\boldsymbol{H}^s, \boldsymbol{H}^e \in \mathbb{R}^{L \times d}$, $\boldsymbol{W}_1 \in \mathbb{R}^{d \times c \times d}$, $\boldsymbol{W}_2 \in \mathbb{R}^{c \times 2d}$, $\boldsymbol{b} \in \mathbb{R}^c$, $\oplus$ refers to concatenation; $\boldsymbol{S} \in \mathbb{R}^{L \times L \times c}$ provides features for all possible token pairs, and $c$ is the feature dimension size.

### 3.2 PLUSFORMER

As illustrated in Section 1, when modeling the interaction between token pairs, the plus-shaped and local interaction should be beneficial. Therefore, we introduce the axis-aware plus-shaped self-attention and position embeddings to conduct plus-shaped interaction, we name this self-attention PlusAttention. Then, we leverage CNN to model local dependencies. We name this whole structure **Plusformer**.

**PlusAttention.** We first apply the self-attention mechanism (Vaswani et al., 2017) horizontally and vertically as follows

$$
\begin{aligned}
\boldsymbol{Z}^h[i,:] &= \text{Attention}(\boldsymbol{S}[i,:]\boldsymbol{W}_h^Q, \boldsymbol{S}[i,:]\boldsymbol{W}_h^K, \boldsymbol{S}[i,:]\boldsymbol{W}_h^V), \\
\boldsymbol{Z}^v[:,j] &= \text{Attention}(\boldsymbol{S}[:,j]\boldsymbol{W}_v^Q, \boldsymbol{S}[:,j]\boldsymbol{W}_v^K, \boldsymbol{S}[:,j]\boldsymbol{W}_v^V), \\
\text{Attention}(\boldsymbol{Q}, \boldsymbol{K}, \boldsymbol{V}) &= \text{softmax}(\frac{\boldsymbol{Q}\boldsymbol{K}^T}{\sqrt{c}})\boldsymbol{V},
\end{aligned}
\tag{3}
$$

where $\boldsymbol{W}_h^Q, \boldsymbol{W}_h^K, \boldsymbol{W}_h^V, \boldsymbol{W}_v^Q, \boldsymbol{W}_v^K, \boldsymbol{W}_v^V \in \mathbb{R}^{c \times c}, \boldsymbol{Z}^h, \boldsymbol{Z}^v \in \mathbb{R}^{L \times L \times c}$. After the self-attention, we use the following method to merge $\boldsymbol{Z}^h, \boldsymbol{Z}^v$

$$
\boldsymbol{S}' = \text{MLP}(\boldsymbol{Z}_h \oplus \boldsymbol{Z}_v),
\tag{4}
$$

where $\boldsymbol{S}' \in \mathbb{R}^{L \times L \times c}$. We make the plus-shaped self-attention axis-aware by using two groups of attention parameters and using concatenation instead of an addition to merge $\boldsymbol{Z}^h, \boldsymbol{Z}^v$.

**Position Embeddings.** Although the model should be able to distinguish between horizontal and vertical directions through axis-aware plus-shaped attention, it still lacks the sense of distances between token pairs and the area the token pair locates. Therefore, we utilize two kinds of position embeddings to enable the model with these abilities.

- **Rotary Position Embedding (RoPE)** (Su et al., 2021) can encode the relative distance between two token pairs. It is utilized in both horizontal and vertical self-attention.
- **Triangle position embedding** is incorporated to mark the position of token pairs in the feature map, which means cells in the upper and lower triangles will use different position embeddings. It adds to $\boldsymbol{S}$ in Eq.(3) before Attention.

**CNN Layer.** After the PlusAttention, we apply CNN with kernel size $3 \times 3$ on the $\boldsymbol{S}'$ to help the model exploit the local dependency between neighboring token pairs. The formulation is as follows

$$
\boldsymbol{S}'' = \text{Conv}(\sigma(\text{Conv}(\boldsymbol{S}')))
\tag{5}
$$

where $\boldsymbol{S}'' \in \mathbb{R}^{L \times L \times c}$, and $\sigma$ is the activation function; and the bias term of CNN is not used to avoid result inconsistencies for a sample when it is in batches of different lengths.

The Plusformer layer will be repeatedly used to interact fully between token pairs. Layer normalization (Ba et al., 2016) is ignored in the formulation for brevity.

### 3.3 LOSS FUNCTION

Finally, we get final scores as follows

$$
\hat{\boldsymbol{Y}}_{\mathcal{S}}, \hat{\boldsymbol{Y}}_{\mathcal{R}} = \text{Sigmoid}(\hat{\boldsymbol{Y}}[:,:,:|\mathcal{S}|]), \hat{\boldsymbol{Y}}[:,:,|\mathcal{S}|:], \qquad \hat{\boldsymbol{Y}} = \text{MLP}(\boldsymbol{S}'' + \boldsymbol{S}),
\tag{6}
$$

where $\hat{\boldsymbol{Y}} \in \mathbb{R}^{L \times L \times (|\mathcal{S}| + |\mathcal{R}| + 1)}$ ; $\hat{\boldsymbol{Y}}_{\mathcal{S}} \in \mathbb{R}^{L \times L \times |\mathcal{S}|}, \hat{\boldsymbol{Y}}_{\mathcal{R}} \in \mathbb{R}^{L \times L \times (|\mathcal{R}| + 1)}$ are scores for span extraction and relational extraction, respectively. The $+1$ in $(|\mathcal{R}| + 1)$ is because we use the adaptive thresholding loss (ATL) from (Zhou et al., 2021) to avoid a global threshold in relational extraction.

For the span extraction, we use the binary cross-entropy (BCE) loss as follows

$$
\mathcal{L}_1 = -\sum_{i,j=1}^{L} \sum_{r=1}^{|\mathcal{S}|} [\boldsymbol{Y}[i,j,r]\log(\hat{\boldsymbol{Y}}[i,j,r]) + (1 - \boldsymbol{Y}[i,j,r])\log(1 - \hat{\boldsymbol{Y}}[i,j,r])]
\tag{7}
$$

For the relational extraction, we utilize the ATL as follows

$$
\begin{aligned}
\mathcal{L}_2 = -\sum_{i,j=1}^{L} \sum_{r \in \mathcal{P}_T} \log\left(\frac{\exp(\hat{\boldsymbol{Y}}_{\mathcal{R}}[i,j,r])}{\sum_{r' \in \mathcal{P}_T \cup \{\text{TH}\}} \exp(\hat{\boldsymbol{Y}}_{\mathcal{R}}[i,j,r'])}\right) \\
-\log\left(\frac{\exp(\hat{\boldsymbol{Y}}_{\mathcal{R}}[i,j,|\mathcal{R}|+1])}{\sum_{r' \in \mathcal{N}^T \cup \{\text{TH}\}} \exp(\hat{\boldsymbol{Y}}_{\mathcal{R}}[i,j,r'])}\right)
\end{aligned}
\tag{8}
$$

where $\mathcal{P}_T$ and $\mathcal{N}_T$ denote the positive and negative classes, $\hat{\boldsymbol{Y}}_{\mathcal{R}}[:,:,|\mathcal{R}|+1]$ is the score for the threshold class TH. Only token pairs with scores higher than their corresponding adaptive thresholds are considered when decoding. We do not use ATL loss for span extraction because we need to sort span scores when decoding spans. The total loss $\mathcal{L} = \mathcal{L}_1 + \mathcal{L}_2$ is used for optimization.

Table 1: Overall F1 on single IE tasks. Results of UTC-IE are the average of 5 runs, and the subscript means the standard deviation (e.g., $93.45_{24}$ means $93.45\pm0.24$). Datasets marked as * have nested entities. Results marked as † are from Yan et al. (2022). ⋆ means results from their Github repo or our reproduction. ♣ means that the UTC-IE without Plusformer surpasses previous SOTA performance.

| *Named Entity Recognition* | CoNLL03 | OntoNotes | ACE04* | ACE05-Ent* | GENIA* |
|---|---|---|---|---|---|
| BART-NER (Yan et al., 2021a) | 93.24 | 90.38 | 86.84 | 84.74 | 78.93 |
| TANL (Paolini et al., 2021) | 91.7 | 89.8 | - | 84.9 | 76.4 |
| $W^2$NER (Li et al., 2022) | 93.07 | 90.50 | $87.43^\dagger$ | $86.77^\dagger$ | $80.32^\dagger$ |
| UIE (Lu et al., 2022) | 92.99 | - | 86.89 | 85.78 | - |
| BS (Zhu & Li, 2022) | $93.39^\star_9$ | $91.51^\star_7$ | $87.08^\dagger$ | $87.20^\dagger$ | - |
| CNN-NER (Yan et al., 2022) | - | - | $87.31^\dagger$ | $87.42^\dagger$ | $80.33^\dagger$ |
| **UTC-IE** | $\mathbf{93.45}_{24}$ | $\mathbf{91.77}_5$ | $\mathbf{87.54}_{33}$ | $\mathbf{87.75}_{35}$ | $\mathbf{80.45}_{22}$ |
| - Plusformer | $92.98_{13}$ | $91.37_5$ | $86.51_{23}$ | $86.59_{20}$ | $79.34_{17}$ |

| *Relation Extraction* | ACE05-R$_{bert}$ | | ACE05-R$_{albert}$ | | SciERC | |
|---|---|---|---|---|---|---|
| | Ent. | Rel. | Ent. | Rel. | Ent. | Rel. |
| TANL (Paolini et al., 2021) | - | - | 88.9 | 63.7 | - | - |
| PURE (Zhong & Chen, 2021) | 88.7 | 63.9 | 89.7 | 65.6 | 66.6 | 35.6 |
| PFN (Yan et al., 2021b) | - | - | 89.0 | 66.8 | $67.2^\star_{67}$ | $37.6^\star_{99}$ |
| UIE (Lu et al., 2022) | - | - | - | 66.06 | - | 36.53 |
| **UTC-IE** | $\mathbf{88.82}_{12}$ | $\mathbf{64.94}_{33}$ | $\mathbf{89.87}_{15}$ | $\mathbf{67.79}_{45}$ | $\mathbf{69.03}_{45}$ | $\mathbf{38.77}_{96}$ |
| -Plusformer | $88.50_{19}$ | $63.34_{72}$ | $89.80^\clubsuit_{23}$ | $66.21_{87}$ | $68.05^\clubsuit_{63}$ | $37.12_{40}$ |

| *Symmetric Relation Extraction* | ACE05-R$^+$ | | SciERC$^+$ | |
|---|---|---|---|---|
| | Ent. | Rel. | Ent. | Rel. |
| UniRE (Wang et al., 2021) | 88.8 | 64.3 | 68.4 | 36.9 |
| PL-Marker (Ye et al., 2022) | 89.8 | 66.5 | 69.9 | 41.6 |
| **UTC-IE** | $\mathbf{90.16}_{21}$ | $\mathbf{67.47}_{74}$ | $\mathbf{69.95}_{41}$ | $\mathbf{42.51}_{42}$ |
| - Plusformer | $88.98_{29}$ | $64.58_{65}$ | $68.78_{63}$ | $39.51_{56}$ |

| *Event Extraction* | ACE05-E | | ACE05-E+ | | ERE-EN | |
|---|---|---|---|---|---|---|
| | Trig. | Arg. | Trig. | Arg. | Trig. | Arg. |
| TANL (Paolini et al., 2021) | 68.4 | 47.6 | - | - | - | - |
| TEXT2EVENT (Lu et al., 2021) | 71.9 | 53.8 | 71.8 | 54.4 | 59.4 | 48.3 |
| UIE (Lu et al., 2022) | - | - | 73.36 | 54.79 | - | - |
| DEGREE (Hsu et al., 2022) | 73.3 | 55.8 | 70.9 | 56.3 | 57.1 | 49.6 |
| **UTC-IE** | $\mathbf{73.46}_{99}$ | $\mathbf{56.51}_{53}$ | $\mathbf{73.44}_{55}$ | $\mathbf{57.68}_{78}$ | $\mathbf{60.20}_{94}$ | $\mathbf{52.51}_{95}$ |
| - Plusformer | $72.88_{78}$ | $55.41_{99}$ | $72.92_{94}$ | $56.63^\clubsuit_{89}$ | $59.28_{77}$ | $51.33^\clubsuit_{99}$ |

# 4 EXPERIMENTS

## 4.1 EXPERIMENTAL SETTINGS

We conduct experiments on 10 datasets across three IE tasks, including NER, RE, and EE, and on 2 joint IE datasets. We evaluate NER task with CoNLL03 (Sang & Meulder, 2003) and OntoNotes (Pradhan et al., 2013) on flat NER, and with ACE04 (Doddington et al., 2004), ACE05-Ent (Walker et al., 2006) and GENIA (Kim et al., 2003) on nested NER. As for relation extraction, we use ACE05-R (Walker et al., 2006) and SciERC (Luan et al., 2018). Since Wang et al. (2021) and Ye et al. (2022) consider symmetric relations, which shall massively influence the performance, we name this scenario Symmetric RE with datasets ACE05-R$^+$ and SciERC$^+$. For event extraction, we follow Lin et al. (2020) to perform experiments on three datasets, ACE05-E, ACE05-E+ (Doddington et al., 2004) and ERE-EN (Song et al., 2015). And for joint IE, we test on ACE05-E+ and ERE-EN. Statistics of all these datasets, implementation details, evaluation metrics, baselines, pre-trained language models and hyper-parameters are described in Appendix C.

Table 2: Results on joint IE. UTC-IE$_{single}$ shows results by separately trained model on NER, RE and EE, while UTC-IE$_{joint}$ shows results by jointly trained model. ♣ means that UTC-IE without Plusformer surpasses previous SOTA performance.

| Joint IE | ACE05-E+ | | | | ERE-EN | | | |
|---|---|---|---|---|---|---|---|---|
| | Ent. | Rel. | Trig. | Arg. | Ent. | Rel. | Trig. | Arg. |
| OneIE (Lin et al., 2020) | 89.6 | 58.6 | 72.8 | 54.8 | 87.0 | 53.2 | 57.0 | 46.5 |
| FourIE (Nguyen et al., 2021) | 91.1 | 63.6 | 73.3 | 57.5 | **87.4** | 56.1 | **57.9** | 48.6 |
| **UTC-IE$_{single}$** | $91.37_{10}$ | $65.00_{49}$ | $71.98_{65}$ | $56.01_{76}$ | $86.35_{48}$ | $55.57_{92}$ | $57.01_{39}$ | $48.29_{60}$ |
| **UTC-IE$_{joint}$** | $\textbf{91.48}_{20}$ | $\textbf{65.54}_{90}$ | $\textbf{73.63}_{47}$ | $\textbf{57.62}_{30}$ | $87.30_{18}$ | $\textbf{56.92}_{90}$ | $57.88_{98}$ | $\textbf{50.91}_{93}$ |
| - Plusformer | $90.72_{30}$ | $62.94_{75}$ | $72.99_{62}$ | $55.68_{74}$ | $86.94_{13}$ | $54.28_{94}$ | $57.79_{83}$ | $48.72_{51}^{♣}$ |

## 4.2 RESULTS ON SINGLE IE TASKS

In this section, we report the UTC-IE performance in each single IE task. Results are shown in Table 1[4]. The table shows that UTC-IE exceeds the previous SOTA models on all IE tasks. Particularly, UTC-IE averagely improves the entity F1 of NER, the entity F1 and relation F1 of RE, the entity F1 and the relation F1 of symmetric RE for +0.18, +0.71, +1.07, +0.21, +0.94, respectively. And for the EE tasks, UTC-IE increases the trigger F1, argument F1 for +0.35 and +1.67. We highlight that our model is significantly helpful for relational extraction, such as relation extraction and argument extraction, which proves the effectiveness of interaction between token pairs. Although the performance increment of span extraction is not as significant as that of relational extraction, UTC-IE consistently improves on various span extraction tasks.

Besides, we also test UTC-IE without Plusformer. Surprisingly, this simple model surpasses previous SOTA models on four results marked with ♣, which proves the effectiveness of the task decomposition. The comparison between models with and without Plusformer clearly shows that Plusformer is effective in all tested datasets, and the performance improvement ranges from +0.40 (on OntoNotes) to +3.00 (on SciERC$^+$). Notably, the average performance gain of adding Plusformer on symmetric RE (+2.06) is more remarkable than that on RE (+1.03). We presume this is because the interaction between token pairs are more beneficial for symmetric relations.

## 4.3 RESULTS ON JOINT IE TASK

Multi-task learning has be proven to be useful in the IE area (Lin et al., 2020; Nguyen et al., 2021). Since UTC-IE unifies all IE tasks into a token-pair classification scenario, it is natural to test whether one UTC-IE model can benefit from jointly learning all IE tasks. In Table 2, the performance of UTC-IE$_{single}$ is from the entity F1 of NER, relation F1 of RE, trigger F1 of EE and argument F1 of EE, respectively. Based on the comparison between UTC-IE$_{single}$ and UTC-IE$_{joint}$, it is obvious that jointly learning these three tasks consistently improves performance in the 2 joint IE datasets.

Moreover, UTC-IE$_{joint}$ outperforms previous SOTA joint IE models in Table 2, the average performance enhancement is +0.69 in ACE05-E+ and +0.75 in ERE-EN. Specifically, UTC-IE$_{joint}$ increases the average performance of relational extraction by +1.30. Thusly, through unifying different IE tasks through our task decomposition, Plusformer can enjoy the benefit of multi-tasking learning, and achieve better performance than previous SOTA models.

## 4.4 SPEED COMPARISON

To get a sense of the speed superiority of UTC-IE, we compare the inference speed of UTC-IE with previous unified models and task-specific SOTA models. The former comparison is presented in Table 3 and the latter locates in the Appendix E. Compared with the generative UIE (Lu et al., 2022), UTC-IE improves F1 from 0.46 (on CoNLL03) to 2.89 (on ACE05-E+), and obtains one order magnitude of speed boost. Compared with OneIE (Lin et al., 2020), UTC-IE fundamentally enhances the performance for relational extractions (e.g., Rel. and Arg.) with an average of 4.47 F1 increment in joint IE. At the same time, UTC-IE is one order of magnitude faster than OneIE. In

---

[4]The precision and recall for UTC-IE in these datasets can be found in Table 8 in Appendix.

Table 3: The F1 and efficiency comparison with UIE and OneIE. "Ent.", "Rel." and "Arg." denote F1 of corresponding test sets. "Speed" is measured in "sentence/s" on inference procedure. The improvement shows the changes in performance and speed.

| *Single IE* | CoNLL03 | | ACE05-Ent | | ACE05-R$_{albert}$ | | SciERC | | ACE05-E+ | |
|---|---|---|---|---|---|---|---|---|---|---|
| | Ent. | Speed | Ent. | Speed | Rel. | Speed | Rel. | Speed | Arg. | Speed |
| UIE (Lu et al., 2022) | 92.99 | 14.5 | 85.78 | 8.6 | 66.06 | 11.4 | 36.53 | 8.7 | 54.79 | 4.0 |
| **UTC-IE** | 93.45 | 210.3 | 87.75 | 304.3 | 67.79 | 85.4 | 38.77 | 165.7 | 57.68 | 88.1 |
| Improvement | **+0.46** | **x14.5** | **+1.97** | **x35.4** | **+1.73** | **x7.5** | **+2.24** | **x19.0** | **+2.89** | **x22.0** |

| *Joint IE* | ACE05-E+ | | | | | ERE-EN | | | | |
|---|---|---|---|---|---|---|---|---|---|---|
| | Ent. | Rel. | Trig. | Arg. | Speed | Ent. | Rel. | Trig. | Arg. | Speed |
| OneIE (Lin et al., 2020) | 89.6 | 58.6 | 72.8 | 54.8 | 4.8 | 87.0 | 53.2 | 57.0 | 46.5 | 13.5 |
| **UTC-IE** | 91.48 | 65.54 | 73.63 | 57.62 | 121.6 | 87.30 | 56.92 | 57.88 | 50.91 | 153.5 |
| Improvement | **+1.88** | **+6.94** | **+0.83** | **+2.82** | **x25.3** | **+0.30** | **+3.72** | **+0.88** | **+4.41** | **x11.4** |

Table 4: Ablation studies in the NER, RE and EE datasets. CNN-IE is similar to UTC-IE except that it is deprived of the PlusAttention. Underlines mean the most dropped factor. ♣ means that the CNN-IE surpasses previous SOTA performance.

| | ACE05-Ent | ACE05-R$_{bert}$ | | ACE05-E+ | |
|---|---|---|---|---|---|
| | Ent. | Ent. | Rel. | Trig. | Arg. |
| **UTC-IE** | $87.75_{35}$ | $88.82_{12}$ | $64.94_{33}$ | $73.44_{55}$ | $57.68_{78}$ |
|   - CNN | $\underline{87.39_{22}}$ | $88.71_{22}$ | $\underline{63.55_{83}}$ | $\underline{72.98_{34}}$ | $\underline{56.74_{99}}$ |
|   - positon embeddings | $87.53_{34}$ | $88.73_{20}$ | $64.29_{56}$ | $73.12_{98}$ | $57.02_{80}$ |
|   - axis-aware | $87.59_{27}$ | $88.79_{19}$ | $63.91_{55}$ | $73.29_{46}$ | $56.87_{98}$ |
| **CNN-IE** | $87.45_{20}^{\clubsuit}$ | $88.70_{16}^{\clubsuit}$ | $64.67_{26}^{\clubsuit}$ | $73.04_{99}$ | $56.97_{63}^{\clubsuit}$ |

a nutshell, compared with previous SOTA models (whether task-specific, unified or joint), UTC-IE achieves substantial performance gain across several datasets with a significant speed boost.

## 4.5 ABLATION STUDY

To analyze the effectiveness of each component in Plusformer, we ablate each of them and list the outcomes in Table 4, and results on more datasets are presented in Appendix F. Besides, we study how many Plusformer layers are suitable in Appendix F.5. Based on the ablation, CNN is the most useful component among all IE tasks. The reason behind this improvement is that once token pairs are organized in the square feature map, the spatial correlations between neighboring token pairs become allusive, and CNN excels at exploiting these local interactions. More comprehensive analysis of CNN in Plusformer locates in Appendix F.1. To deepen our understanding of UTC-IE, we try another variant of Plusformer where the PlusAttention is discarded, and we name this variant **CNN-IE**. The bottom line of Table 4 shows that the CNN-IE model can surpass or approach previous SOTA performance in almost all datasets, which proves the universality of our proposed task formulation.

However, CNN is not a panacea for UTC-IE. From Table 4, removing position embeddings or axis-aware[5] from UTC-IE will lead to an average of 0.39 or 0.44 performance degradation, respectively. Moreover, based on the performance of CNN-IE and UTC-IE, the average performance shrinks from 74.53 to 74.17 if the PlusAttention is deprived of Plusformer, which means the plus-shaped self-attention is a desideratum. In addition, we present some intuitive examples and deeper analysis for position embeddings and axis-aware in Appendix F.3 and F.4.

---

[5]Removing axis-aware means using the same self-attention parameters for both directions and adding $\boldsymbol{Z}^h$ and $\boldsymbol{Z}^v$ instead of concatenation.

## 5 RELATED WORK

Information extraction tasks, which consists of named entity recognition, relation extraction, and event extraction, have long been a fundamental and well-researched task in the natural language processing (NLP) field. Previous researches mainly only focus on one or two tasks. Recently, building joint neural models of unified IE tasks has attracted increasing attention. Some of them incorporate graphs into IE structure. Wadden et al. (2019) propose a unified framework called DYGIE++ to extract entities, relations and events by leveraging span representations via span graph updates. Lin et al. (2020) and Nguyen et al. (2021) extend DYGIE++ by incorporating global features to extract cross-task and cross-instance interactions with multi-task learning. In addition to the graph-based models mentioned above, other studies focus on tackling general IE by generative models. Paolini et al. (2021) construct a framework called TANL, which enhances the generation model using augmented language methods. Moreover, Lu et al. (2022) regard IE task as a text-to-structure generation task, and leveraging prompt mechanism.

We unify all IE tasks as several token-pair classification tasks, which are fundamentally similar to the span-based methods on the IE task, for the start and end tokens can locate a span. Numerous NER studies emerge on span-based models, which are compatible with both flat and nested entities and perform well (Eberts & Ulges, 2020; Yu et al., 2020b; Li et al., 2021; Zhu & Li, 2022). In addition to entities, the span-based method is also used in RE. Some models (Wang et al., 2021; Ye et al., 2022) only leverage span representations to locate entities and simply calculate the interaction between entity pair, while others (Wang et al., 2020; Zhong & Chen, 2021) encode span pair information explicitly to extract relations. With regard to event extraction, as far as we know, there is little work on injecting span information into EE explicitly. Wadden et al. (2019) leveraging span representations on general IE, but their model is complicated and only considers span at the embedding layer without further interaction. Conceptually, Jiang et al. (2020)'s work is similar to ours, but they need a two-stage model to determine the span type and span relations, respectively. Detailed analysis are depicted in Appendix G. Although many span-based IE models exist, they are task-specific and lack interaction between token pairs. Decomposing IE tasks as token-pair classification and conducting interaction between token pairs can uniformly model span-related knowledge and advance SOTA performance.

The key component of Plusformer is the plus-shaped attention mechanism, which can make token pairs interact with each other in an efficient way. A similar structure called Axial Transformers (Ho et al., 2019) is proposed in Computer Vision (CV) field, which is designed to deal with data organized as high-dimension tensors. Tan et al. (2022) incorporate axial attention into relation classification, aiming to improve the performance on two-hop relation. However, CNN was not used in these work, while CNN has been proven to be vital to the IE tasks. Another similar structure named Twin Transformer (Guo et al., 2021) used in CV is very much similar to Plusformer, where they encode pixel of image from row and column sequentially, and leverage CNN on top of them. But the position embeddings, which is important for IE tasks, are not used in the Twin Transformer. Besides, we want to point out that the usage of plus-shaped attention and CNN originates from the reformulation of IE tasks, any other modules which can directly enable interaction between constituent spans of a relation and between adjacent token pairs should be beneficial.

## 6 CONCLUSION

In this paper, we decompose NER, RE and EE tasks into token-pair classifications. Through the decomposition, we unify all IE tasks under the same formulation. After scrutinizing the token-pair feature matrix, we find the adjacent and plus-shaped interactions between token pairs should be informative. Therefore, we propose Plusformer, which uses an axis-aware plus-shaped self-attention followed by CNN layers to help token pairs interact with each other. Experiments in 10 single IE datasets and 2 joint IE datasets all outperform or approach the SOTA performance. Besides, owing to the parallelism of self-attention and CNN, our model's inference speed is substantially faster than previous SOTA models in RE and EE. Lastly, most of the previous IE models limit the interaction in the 1-D sequential dimension, while the reformulation of IE tasks opens a new angle to broaden the communication to the 2-D feature matrix.

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

## A  DISCUSSION ON TASK DECOMPOSITION

In this section, we will discuss two issues of the decomposition. The first is the inconsistency stipulation about the relation decomposition, the second is the false positive issue when decoding relations.

### A.1  THE INCONSISTENCY

As our stipulation in Section 2, if a span $(s, e)$ has an expected span type $t$, both the $\boldsymbol{Y}[s, e, t]$ and $\boldsymbol{Y}[e, s, t]$ are 1. If two spans $(s_1, e_1)$ and $(s_2, e_2)$ have relation $r$, this means the relation should also exist between spans $(s_1, e_1)$ and $(e_2, s_2)$ (the end-to-start version of the span $(s_2, e_2)$), then based on our stipulation on the relation, the $\boldsymbol{Y}[s_1, e_2, r]$ and $\boldsymbol{Y}[e_1, s_2, r]$ should also equal 1, but we only define the $\boldsymbol{Y}[s_1, s_2, r] = 1$ and $\boldsymbol{Y}[e_1, e_2, r] = 1$, this causes an inconsistency between the stipulations. We ignore $\boldsymbol{Y}[s_1, e_2, r]$ and $\boldsymbol{Y}[e_1, s_2, r]$ to make the decoding less cluttered.

### A.2  FALSE POSITIVE RELATION

A potential risk of the decomposition and decoding is that it may cause false positive relations. Given four spans $p_1 = (s_1, e_1), p_2 = (s_2, e_2), p_3 = (s_3, e_3), p_4 = (s_4, s_4)$, if $p_4$ has relation $r$ with $p_1$ and $p_2$, and no relation exist between $p_4$ and $p_3$. Then $\boldsymbol{Y}[s_4, s_1, r] = \boldsymbol{Y}[e_4, e_1, r] = 1$, $\boldsymbol{Y}[s_4, s_2, r] = \boldsymbol{Y}[e_4, e_2, r] = 1$. However, if $s_1 = s_3$, $e_2 = e_3$. Namely, $p_1$ shares start token with $p_3$ and $p_2$ shares end token with $p_3$. Then, based on $\boldsymbol{Y}[s_4, s_1, r] = \boldsymbol{Y}[e_4, e_2, r] = 1$, we get $\boldsymbol{Y}[s_4, s_3, r] = \boldsymbol{Y}[e_4, e_3, r] = 1$,, the decoding process will mistakenly think $p_4$ has relation $r$ with $p_3$. However, this situation should be rare, and none is found in the tested datasets.

## B  DECODING WITH MODEL'S PREDICTIONS

In this section, we will detail the decoding process for models' predictions. The process described in Section 2.2 is not directly applicable to models' predictions since spans may conflict with each other[6]. With prediction score matrix $\hat{\boldsymbol{Y}}_{\mathcal{S}}$ from Eq.(6), we follow previous work (Yu et al., 2020b) to first filter out spans whose scores are less than 0.5; for the left spans, we sort the spans based on their scores, then choose spans in descending order and make sure the span has no boundary clash with chosen spans. For relational extraction, we first decode all spans, then we get a binary matrix

---

[6]All IE tasks forbid span boundary clashes.

$\bar{Y}_{\mathcal{R}} = \hat{Y}_{\mathcal{R}}[:,:,:|R|+1] > \hat{Y}_{\mathcal{R}}[:,:,|R|+1]$, then we pair spans to check whether they form relations. Take two spans $(s_1, e_1)$ and $(s_2, e_2)$ for instance, if $\bar{Y}_{\mathcal{R}}[s_1, s_2, r] = \bar{Y}_{\mathcal{R}}[e_1, e_2, r] = 1$, we claim the first span has relation $r$ with the second span. For the RE task, we pair all entity spans to check if they form relations; for the EE task, we pair the trigger spans and argument spans to check if they form a role relationship; and for the joint IE task, we pair entity spans to check if they form relations, we pair the trigger spans and entity spans (because all argument spans are entity spans) to check if they form a role relationship.

## C  EXPERIMENTAL SETTINGS

In this section, we describe all experimental settings in detail, such as the statistics of datasets, baseline models, and more implementation details.

### C.1  DATASETS

We conduct experiments on 10 single IE datasets and 2 joint IE datasets, and we detail the statistic of all datasets in Table 5.

|  | #Train | #Dev | #Test | #Ents (#Types) | #Rels (#Types) | #Evts (#Types) |
|---|---|---|---|---|---|---|
| CoNLL03 | 14,041 | 3,250 | 3,453 | 35.1k (4) | - | - |
| OntoNotes | 59,924 | 8,528 | 8,262 | 104.2k (18) | - | - |
| ACE04 | 6,297 | 742 | 824 | 27.8k (7) | - | - |
| ACE05-Ent | 7,178 | 960 | 1,051 | 31.7k (7) | - | - |
| GENIA | 15,038 | 1,765 | 1,732 | 56.0k (5) | - | - |
| ACE05-R | 10,051 | 2,424 | 2,050 | 38.3k (7) | 7.1k (6) | - |
| ACE05-R$^+$ | 10,051 | 2,424 | 2,050 | 38.3k (7) | 7.7k (6) | - |
| SciERC | 1,861 | 275 | 551 | 8.1k (6) | 4.6k (7) | - |
| SciERC$^+$ | 1,861 | 275 | 551 | 8.1k (6) | 5.5k (7) | - |
| ACE05-E | 17,172 | 923 | 832 | 34.5k (7) | 5.9k (6) | 5.1k (33;22) |
| ACE05-E+ | 19,204 | 901 | 676 | 54.7k (7) | 8.7k (6) | 5.3k (33;22) |
| ERE-EN | 14,722 | 1,209 | 1,163 | 46.2k (7) | 5.9k (5) | 7.3k (38;21) |

Table 5: Datasets statistics. "#Types" denotes the number of classes. Note that "#Types" in the last column mean (#event types; #role type) pairs. Every block represents datasets of different tasks, which are flat NER, nested NER, RE and EE from top to bottom. For the joint IE setting, the "ACE05-E+" and "ERE-EN" are used. In the RE block, datasets following $^+$ mean that each symmetric relational instance is regarded as two directional instances, leading to more relations.

**Named entity recognition.** We perform experiments on both flat and nested NER benchmarks. In flat NER, we adopt CoNLL03 (Sang & Meulder, 2003) and OntoNotes[7] (Pradhan et al., 2013) datasets. In nested NER, we experiment on ACE04[8] (Doddington et al., 2004), ACE05[9] (Walker et al., 2006) and GENIA (Kim et al., 2003). To distinguish ACE05 dataset used in other tasks, we name ACE05 in named entity recognition as ACE05-Ent. Specifically, we use the same preprocessing and splitting procedure on nested datasets as Yan et al. (2022), for they fix some annotation problems to unify different versions of these datasets and make a strictly fair comparison.

**Relation extraction.** We conduct experiments on two relation extraction datasets, ACE05 (Walker et al., 2006) and SciERC[10] (Luan et al., 2018). The ACE05 dataset, named as ACE05-R in our paper, is collected from various domains, such as newswire and online forums. The SciERC dataset provides entity, coreference and relation annotations from AI conference/workshop proceedings.

---

[7] https://catalog.ldc.upenn.edu/LDC2013T19
[8] https://catalog.ldc.upenn.edu/LDC2005T09
[9] https://catalog.ldc.upenn.edu/LDC2006T06
[10] http://nlp.cs.washington.edu/sciIE/

We follow the data preprocessing in Luan et al. (2019) to split ACE05-R and SciERC into train, dev and test sets.

In typical RE, it is crucial to distinguish which entity comes first (head entity) and which comes next (tail entity). As for symmetric relational instance, the relation exists from both head-to-tail and tail-to-head directions. There are one such relation type in ACE05-R and two in SciERC. Some papers (Wang et al., 2021; Ye et al., 2022) regard each symmetric relational example as two directed relations, while others regard them as one relation. We find that this setting will hugely influence the performance. Therefore, we name this setting **Symmetric Relation Extraction** and name the corresponding datasets ACE05-R$^+$ and SciERC$^+$.

**Event extraction.** We evaluate UTC-IE on two widely used event extraction datasets, ACE2005 (Doddington et al., 2004) and ERE (Song et al., 2015). Following the prior preprocessing step (Wadden et al., 2019; Lin et al., 2020; Lu et al., 2021) on them, we obtain three datasets, ACE05-E, ACE05-E+ and ERE-EN. ACE05-E+ additionally takes relation arguments, pronouns and multi-token event triggers into consideration compared with ACE05-E. We use the same train/dev/test split as Lu et al. (2021) for all datasets to ensure a fair comparison. Furthermore, we still use ACE05-E+ and ERE-EN on joint IE, for they have annotations on all IE tasks.

## C.2 BASELINES

**TANL** (Paolini et al., 2021) and **UIE** (Lu et al., 2022) are both unified information extraction models in the generative way, with different input and output formats. TANL uses T5-base as the backbone model, while UIE uses T5-large. We compare our model with them in every IE task. For TANL, we report single-task results for our model is trained under each task. For UIE, we report results with pre-training, which have better performance. In addition to these two baselines, each task also compares with a series of recently proposed task-specific methods as follows.

**CNN-IE** is the baseline model we design to prove the necessity of PlusAttention. The only difference between CNN-IE and UTC-IE is the former ignores the PlusAttention in Figure 2. We tune the number of CNN layers in CNN-IE from 2 to 6, and the best results are reported.

**Named entity recognition.** We compare our model's performance on NER with several recently proposed NER methods.

- **BART-NER** (Yan et al., 2021a) formulates unified NER model as entity span sequence generation task. They use BART-large as the pre-trained model.
- **W$^2$NER** (Li et al., 2022) formulates unified NER model as word-to-word classification task. The model employs BioBERT on GENIA and BERT-large on other datasets.
- **BS** (Zhu & Li, 2022): authors use span-based NER model as baseline and propose boundary smoothing as a regularization technique to improve model performance. It leverages RoBERTa-base as the base encoder.
- **CNN-NER** (Yan et al., 2022) utilizes CNN to model local spatial correlations between spans and surpass recently proposed methods on nested NER. We report results using RoBERTa-base model.

**Relation extraction.** For relation extraction, we compare our model with several SOTA models.

- **UniRE** (Wang et al., 2021) jointly extracts entities and relations using a table containing all word pairs.
- **PURE** (Zhong & Chen, 2021) adopts a pipeline approach to solve NER and RE independently, using distinct contextual representations for entities and relations.
- **PFN** (Yan et al., 2021b) claims that some information should be shared between named entity recognition and relation extraction, while other information should be independent. They propose PFN to model two-way interaction (partition and filter) between two tasks.
- **PL-Marker** (Ye et al., 2022): authors consider interactions between spans and propose PL-Marker by strategically packing the markers in the encoder.

Table 6: Overall pre-trained model on all IE baselines. Abbreviations before "-" denote pre-trained model names. Specifically, "BA" means BART, "BE" means BERT, "RoB" means RoBERTa, "ALB" means ALBERT, "DeB" means DeBERTa. The letters after "-" means the size of the model, such as base model ("b"), large model ("l"), xx-large model ("xxl"). The number of parameters of each pre-trained model is as follows: BE-b (110M), BE-l (340M), RoB-b (125M), ALB-xxl (233M), DeB-l (390M), T5-b (220M), T5-l (770M), BA-l (406M).

| *Named Entity Recognition* | CoNLL03 | OntoNotes | ACE04* | ACE05-Ent* | GENIA* |
|---|---|---|---|---|---|
| BART-NER (Yan et al., 2021a) | BA-l | BA-l | BA-l | BA-l | BA-l |
| TANL (Paolini et al., 2021) | T5-b | T5-b | - | T5-b | T5-b |
| W$^2$NER (Li et al., 2022) | BE-l | BE-l | BE-l | BE-l | BioBERT |
| UIE (Lu et al., 2022) | T5-l | - | T5-l | T5-l | - |
| BS (Zhu & Li, 2022) | RoB-b | RoB-b | RoB-b | RoB-b | - |
| CNN-NER (Yan et al., 2022) | - | - | RoB-b | RoB-b | BioBERT |
| **UTC-IE** | RoB-b | RoB-b | RoB-b | RoB-b | BioBERT |

| *Relation Extraction* | ACE05-R$_{bert}$ | | ACE05-R$_{albert}$ | | SciERC |
|---|---|---|---|---|---|
| TANL (Paolini et al., 2021) | - | | T5-b | | - |
| PURE (Zhong & Chen, 2021) | BE-b | | ALB-xxl | | SciBERT |
| PFN (Yan et al., 2021b) | BE-b | | ALB-xxl | | SciBERT |
| UIE (Lu et al., 2022) | - | | T5-l | | T5-l |
| **UTC-IE** | BE-b | | ALB-xxl | | SciBERT |

| *Symmetric Relation Extraction* | ACE05-R$^+$ | | SciERC$^+$ |
|---|---|---|---|
| UniRE (Wang et al., 2021) | BE-b | | SciBERT |
| PL-Marker (Ye et al., 2022) | BE-b | | SciBERT |
| **UTC-IE** | BE-b | | SciBERT |

| *Event Extraction* | ACE05-E | ACE05-E+ | ERE-EN |
|---|---|---|---|
| TANL (Paolini et al., 2021) | T5-b | - | - |
| TEXT2EVENT (Lu et al., 2021) | T5-l | T5-l | T5-l |
| UIE (Lu et al., 2022) | - | T5-l | - |
| DEGREE (Hsu et al., 2022) | BA-l | BA-l | BA-l |
| **UTC-IE** | DeB-l | DeB-l | DeB-l |

| *Joint IE* | ACE05-E+ | | ERE-EN |
|---|---|---|---|
| OneIE (Lin et al., 2020) | BE-l | | BE-l |
| FourIE (Nguyen et al., 2021) | BE-l | | BE-l |
| **UTC-IE** | BE-l | | BE-l |

Previous models mentioned above use different RE datasets. Specifically, UniRE and PL-Marker regard symmetric relations as two directed relations, while other work does not. Besides, these two models utilize cross-sentence context.

**Event extraction.** Generative methods are popular in recently proposed event extraction papers.

- **TEXT2EVENT** (Lu et al., 2021) is a sequence-to-structure model which outputs a tree-like event structure with an given input sentence. The model uses T5-large as the base model.
- **DEGREE** (Hsu et al., 2022) leverages manually designed prompts to generate event records in natural language. We report the end-to-end performance of DEGREE instead of the pipeline way. The model leverages BART-large as encoder-decoder.

**Joint IE.** There are only two previous models that consider the joint IE in ACE05-E and ERE-EN datasets.

- **OneIE** (Lin et al., 2020) proposes an end-to-end IE model, which employs global features and type dependency constraint at decoding step.
- **FourIE** (Nguyen et al., 2021) further improves the model by incorporating interaction dependency on representation level and label level.

For a fair comparison, we list the pre-trained model used for all baselines and our model on every IE dataset in Table 6. When choosing our pre-trained language model in different IE tasks' datasets, we pick the same pre-trained model as the most recently published papers, such as BioBERT for GENIA and RoBERTa-base for other NER datasets. For RE and joint IE tasks, we choose the same pre-trained model as previous work. For tasks where previous work applied a generative pre-trained model, we choose pre-trained model that has a similar size. For example, in event extraction, we use DeBERTa-large, whose number of parameters is 390M, which is closest to BART-large and T5-large used by previous EE papers.

### C.3 EVALUATION METRICS

We report micro-F1 on all tasks:

- **Entity:** an entity is correct if its entity type and span offsets match a golden entity. We use "Ent." to represent entity F1 through all tables.
- **Relation:** a relation is correct if its type and its head and tail entities are correct, and the offsets and type of entities should also match the golden instance. We use "Rel." to represent relation F1 through all tables.
- **Event trigger:** a trigger is correct if its span offset and event type is correct. We use "Trig." to represent trigger F1 through all tables.
- **Event argument:** an argument is correct if its span offset, event type and role type all match the ground truth. We use "Arg." to represent argument F1 through all tables.

### C.4 HYPER-PARAMETERS

The detailed hyper-parameters used in each dataset are listed in Table 7. AdamW optimizer (Loshchilov & Hutter, 2019) with weight decay 1e-2 for all datasets. Experiments are conducted five times with five different random seeds. We report the performance on test sets based on the model which achieves the best dev results in each dataset. For NER, the best results are calculated by the entity F1; for RE, the best results are calculated by the sum of entity F1 and relation F1; for EE, the best results are calculated by the best argument F1; for joint IE, the best results are calculated by the sum of relation F1 and trigger F1.

## D COMPLETE RESULTS

We present the complete results of UTC-IE and that without Plusformer in Table 8.

Table 7: The hyper-parameters used in each dataset.

| *Named Entity Recognition* | CoNLL03 | OntoNotes | ACE04* | ACE05-Ent* | GENIA* |
|---|---|---|---|---|---|
| # Epochs | 30 | 10 | 50 | 50 | 5 |
| Learning Rate | 1e-5 | 1e-5 | 2e-5 | 2e-5 | 7e-6 |
| Batch Size | 12 | 12 | 48 | 48 | 8 |
| # Plusformer Layers | 2 | 1 | 2 | 2 | 2 |
| Biaffine Dimension $d$ | 200 | 200 | 200 | 200 | 200 |
| Feature Dimension $c$ | 32 | 100 | 100 | 100 | 100 |
| Warmup Ratio | 0.1 | 0.1 | 0.2 | 0.2 | 0.1 |
| *Relation Extraction* | ACE05-$R_{bert}$ | ACE05-$R_{albert}$ | SciERC | ACE05-$R^+$ | SciERC$^+$ |
| # Epochs | 100 | 100 | 70 | 50 | 100 |
| Learning Rate | 3e-5 | 3e-5 | 3e-5 | 3e-5 | 3e-5 |
| Batch Size | 32 | 32 | 16 | 32 | 16 |
| # Plusformer Layers | 3 | 3 | 3 | 3 | 3 |
| Biaffine Dimension $d$ | 200 | 200 | 200 | 200 | 200 |
| Feature Dimension $c$ | 200 | 200 | 200 | 200 | 200 |
| Warmup Ratio | 0.1 | 0.1 | 0.1 | 0.1 | 0.1 |
| *Event Extraction* | ACE05-E | | ACE05-E+ | | ERE-EN |
| # Epochs | 70 | | 70 | | 70 |
| Learning Rate | 1e-5 | | 1e-5 | | 1e-5 |
| Batch Size | 32 | | 32 | | 32 |
| # Plusformer Layers | 3 | | 3 | | 3 |
| Biaffine Dimension $d$ | 300 | | 300 | | 300 |
| Feature Dimension $c$ | 150 | | 150 | | 150 |
| Warmup Ratio | 0.1 | | 0.1 | | 0.1 |
| *Joint IE* | ACE05-E+ | | | ERE-EN | |
| # Epochs | 70 | | | 30 | |
| Learning Rate | 1e-5 | | | 3e-5 | |
| Batch Size | 12 | | | 12 | |
| # Plusformer Layers | 3 | | | 3 | |
| Biaffine Dimension $d$ | 300 | | | 300 | |
| Feature Dimension $c$ | 150 | | | 150 | |
| Warmup Ratio | 0.1 | | | 0.1 | |

Table 8: Completed results for precision (P), recall (R) and F1 (F) of UTC-IE on different tasks. Bold results represent the most improved metrics on UTC-IE without Plusformer between precision and recall.

| Named Entity Extraction | CoNLL03 | | | OntoNotes | | | ACE04* | | | ACE05-Ent* | | | GENIA* | | |
|---|---|---|---|---|---|---|---|---|---|---|---|---|---|---|---|
| | P | R | F | P | R | F | P | R | F | P | R | F | P | R | F |
| UTC-IE | 93.4 | **93.6** | 93.5 | **91.7** | 91.9 | 91.8 | 87.3 | **87.7** | 87.5 | 86.8 | **88.8** | 87.8 | 81.6 | **79.4** | 80.5 |
| - Plusformer | 93.0 | 93.0 | 93.0 | 91.0 | 91.8 | 91.4 | 86.8 | 86.2 | 86.5 | 85.8 | 87.4 | 86.6 | 81.6 | 77.2 | 79.3 |

| Relation Extraction | ACE05-$R_{bert}$ | | | | | | ACE05-$R_{albert}$ | | | | | | SciERC | | | | | |
|---|---|---|---|---|---|---|---|---|---|---|---|---|---|---|---|---|---|---|
| | Ent. | | | Rel. | | | Ent. | | | Rel. | | | Ent. | | | Rel. | | |
| | P | R | F | P | R | F | P | R | F | P | R | F | P | R | F | P | R | F |
| UTC-IE | **88.7** | 89.0 | 88.8 | **70.1** | 60.5 | 64.9 | 89.3 | **90.5** | 89.9 | 70.4 | **65.4** | 67.8 | 68.0 | **70.1** | 69.0 | 43.6 | **34.9** | 38.8 |
| - Plusformer | 88.1 | 88.9 | 88.5 | 66.7 | 58.4 | 63.3 | 89.7 | 90.0 | 89.8 | 70.2 | 62.7 | 66.2 | 67.4 | 68.9 | 68.1 | 43.0 | 32.7 | 37.1 |

| Symmetric Relation Extraction | ACE05-$R^+$ | | | | | | SciERC$^+$ | | | | | |
|---|---|---|---|---|---|---|---|---|---|---|---|---|
| | Ent. | | | Rel. | | | Ent. | | | Rel. | | |
| | P | R | F | P | R | F | P | R | F | P | R | F |
| UTC-IE | **90.0** | 90.5 | 90.2 | **69.3** | 65.8 | 67.5 | 68.5 | **71.5** | 70.0 | 45.7 | **39.8** | 42.5 |
| - Plusformer | 87.5 | 90.5 | 89.0 | 67.3 | 64.0 | 64.6 | 68.0 | 69.6 | 68.8 | 43.5 | 36.2 | 39.5 |

| Event Extraction | ACE05-E | | | | | | ACE05-E+ | | | | | | ERE-EN | | | | | |
|---|---|---|---|---|---|---|---|---|---|---|---|---|---|---|---|---|---|---|
| | Ent. | | | Rel. | | | Ent. | | | Rel. | | | Ent. | | | Rel. | | |
| | P | R | F | P | R | F | P | R | F | P | R | F | P | R | F | P | R | F |
| UTC-IE | 70.9 | 76.2 | 73.5 | **55.5** | 57.6 | 56.5 | 70.8 | **76.1** | 73.4 | **57.8** | 57.6 | 57.7 | **58.1** | 62.5 | 60.2 | **54.5** | 50.7 | 52.5 |
| - Plusformer | 70.1 | 76.0 | 72.9 | 52.5 | 58.7 | 55.4 | 70.5 | 75.5 | 72.9 | 55.6 | 57.7 | 56.6 | 56.0 | 63.0 | 59.3 | 52.3 | 50.3 | 51.3 |

# E    SPEED COMPARISION

We test the speed of other models through their released code. For models, such as OneIE (Lin et al., 2020), DEGREE (Hsu et al., 2022) and PL-Marker[11] (Ye et al., 2022), they also released a trained model along with their code, and we used their released model to test the inference speed. For UIE (Lu et al., 2022) and BS (Zhu & Li, 2022), we trained a model with their code. The speed test is conducted in one RTX 3090 GPU and the batch size is set as 32 for all models (if the model goes out of memory, we choose the largest batch size that can accommodate the GPU); the test corpus is the test set of each datasets. The speed is measured by the number of sentences in the test set divided by the number of seconds that elapsed. And each inference is repeated three times, the average speed is reported.

The speed comparison can be roughly categorized into two kinds. The first kind is the comparison with previous universal IE models, namely OneIE (Lin et al., 2020) and UIE (Lu et al., 2022), and results are depicted in Table 3. Compared with UIE in five chosen datasets, UTC-IE is x19.7 faster and improves performance by 1.86 averagely. Besides, for the joint IE task, UTC-IE is 18.4 times faster than OneIE and improves performance by 2.72 on average. The second kind is the comparison between UTC-IE and SOTA models targeted for each IE task, and results are presented in Table 9. Compared with previous SOTA models, the average performance increments for entity F1, relation F1 and argument F1 are 0.31, 0.94 and 2.15. In the meantime, UTC-IE speeds up for x1.0, x5.5 and x101.9 averagely.

In short, using UTC-IE for IE tasks can not only substantially enhance the performance in most cases, but also significantly speed up the inference speed in almost all datasets.

---

[11]PL-Marker used a two-stage pipeline to conduct prediction. Therefore, the time is measured by the total seconds elapse to finish two stages.

Table 9: The F1 and inference time comparison on UTC-IE and currently SOTA models on each IE task. "Ent.", "Rel." and "Arg." denote F1 of corresponding test sets. "Speed" is measured in "sentence/s" on inference procedure. Improvement shows the changes in performance and speed.

| Named Entity Recognition | CoNLL03 | | ACE05-Ent | |
| --- | --- | --- | --- | --- |
| | Ent. | Speed | Ent. | Speed |
| BS (Zhu & Li, 2022) | 93.39 | 265.6 | 87.20 | 355.4 |
| **UTC-IE** | 93.45 | 285.3 | 87.75 | 344.3 |
| Improvement | **+0.06** | **x1.1** | **+0.55** | x1.0 |

| Symmetric Relation Extraction | ACE05-R$^+$ | | SciERC$^+$ | |
| --- | --- | --- | --- | --- |
| | Rel. | Speed | Rel. | Speed |
| PL-Marker (Ye et al., 2022) | 66.5 | 30.1 | 41.6 | 26.0 |
| **UTC-IE** | 67.47 | 173.8 | 42.51 | 134.7 |
| Improvement | **+0.97** | **x5.8** | **+0.91** | **x5.2** |

| Event Extraction | ACE05-E+ | | ERE-EN | |
| --- | --- | --- | --- | --- |
| | Arg. | Speed | Arg. | Speed |
| DEGREE (Hsu et al., 2022) | 56.3 | 0.8 | 49.6 | 1.2 |
| **UTC-IE** | 57.68 | 88.1 | 52.51 | 114.6 |
| Improvement | **+1.38** | **x107.4** | **+2.91** | **x96.3** |

Table 10: Ablation Study for span extraction. Underlines mean the most dropped factor. ♣ means the CNN-IE surpasses previous SOTA performance.

| | CoNLL03 Ent. | ACE05-Ent Ent. | ACE05-R$_{bert}$ Ent. | SciERC Ent. | ACE05-E+ Trig. | ERE-EN Trig. |
| --- | --- | --- | --- | --- | --- | --- |
| **UTC-IE** | $93.45_{24}$ | $87.75_{35}$ | $88.82_{12}$ | $69.03_{45}$ | $73.44_{55}$ | $60.20_{94}$ |
| - CNN | $93.10_{11}$ | $87.39_{22}$ | $88.71_{22}$ | $68.35_{56}$ | $72.98_{34}$ | $58.91_{31}$ |
| - position embeddings | $93.25_{11}$ | $87.53_{34}$ | $88.73_{20}$ | $68.69_{58}$ | $73.12_{98}$ | $59.03_{70}$ |
| - axis-aware | $93.23_{10}$ | $87.59_{27}$ | $88.79_{19}$ | $68.53_{48}$ | $73.29_{46}$ | $59.56_{99}$ |
| **CNN-IE** | $93.32_{16}$ | $87.45^{♣}_{20}$ | $88.70^{♣}_{16}$ | $68.11^{♣}_{71}$ | $73.04_{99}$ | $59.47^{♣}_{63}$ |

# F    ABLATION STUDY

For ablation, we will choose two datasets for each IE task to study the effect of each component in Plusformer. We separately list the performance for span extraction (including entity extraction in NER and RE, trigger extraction in EE) in Table 10 and relational extraction (including relation extraction in RE and argument extraction in EE) in Table 11. Besides, we also study how the performance varies with the change of the number of Plusformer layers in Figure 10.

## F.1    CNN

Based on our ablations in Table 10 and Table 11, the CNN module in Plusformer contributes most to the performance enhancement. To reveal why CNN is so effective in both span extraction and relational extraction, we first present an intuitive example in Figure 3 to show how CNN helps to extract entities and relations in the RE task. Like in Figure 3(a), for NER, the entity $e_2$ can interact with entity $e_1$ and relation $r_1$ through CNN. Besides, for RE, CNN can contribute in two ways. On the one hand, CNN helps the relational token pair to directly gather information from its constituent entities, like the $r_1$ in Figure 3(b). On the other hand, the start-to-start and end-to-end relational token pairs, like two $r_2$ cells, can directly interact with each other through CNN.

To quantitatively present the effectiveness of CNN in UTC-IE, we propose further ablations to show how the distance between the relational token pair and its constituent spans affects the relational F1, and how the distance between start-to-start and end-to-end token pairs affects the relational F1.

Table 11: Ablation Study for relational extraction. Underlines mean the most dropped factor. ♣ means that the CNN-IE surpasses previous SOTA performance.

| | ACE05-R$_{bert}$ Rel. | SciERC Rel. | ACE05-E+ Arg. | ERE-EN Arg. |
|---|---|---|---|---|
| **UTC-IE** | $64.94_{33}$ | $38.77_{96}$ | $57.68_{78}$ | $52.51_{95}$ |
| - CNN | $\underline{63.55_{83}}$ | $\underline{37.56_{83}}$ | $\underline{56.74_{99}}$ | $\underline{51.59_{99}}$ |
| - position embeddings | $64.29_{56}$ | $37.98_{99}$ | $57.02_{80}$ | $52.06_{70}$ |
| - axis-aware | $63.91_{55}$ | $37.76_{83}$ | $56.87_{98}$ | $51.92_{69}$ |
| **CNN-IE** | $64.67_{26}^{\clubsuit}$ | $37.64_{65}^{\clubsuit}$ | $56.97_{63}^{\clubsuit}$ | $51.78_{50}^{\clubsuit}$ |

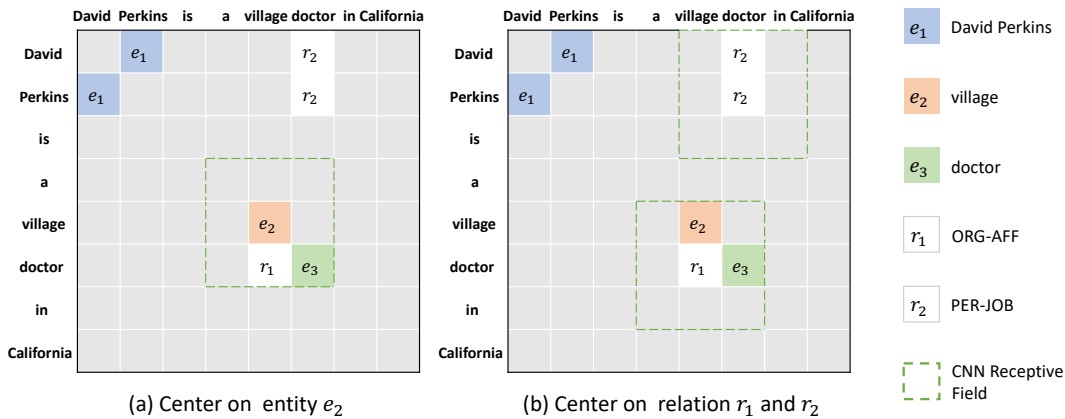

(a) Center on entity $e_2$     (b) Center on relation $r_1$ and $r_2$

Figure 3: An intuitive example of the influence of CNN on span extraction and relation extraction.

Furthermore, we conduct experiments on UTC-IE with different kernel sizes and choose the most proper size.

#### F.1.1 Distance Between the Relational Token Pair and Its Constituent spans VS. Relational F1

In this section, we will show how the relational F1 (relation F1 in RE and argument F1 in EE) will change when the distance between the relational token pair and its constituent spans varies. For two spans $(s_1, e_1)$ and $(s_2, e_2)$ (we ignore their diagonally symmetric counterparts, since they will not affect the calculation here), the span relation from $(s_1, e_1)$ to $(s_2, e_2)$ is represented by two token pairs $(s_1, s_2)$ and $(e_1, e_2)$. The distance between the two token pairs and its constituent spans is calculated as follows

$$d = \max(|s_2 - e_1|, |s_1 - e_2|) + 1, \tag{9}$$

where the distance $d$ is named as "**Span-Rel-Span Distance**", it represents the longest distance between the relational token pairs to their constituent spans. The relation between $d$ and the relational F1 is shown in Figure 4. Without CNN, the performance for extracting relations between nearby constituent spans will drop sightly, while less affected for further ones, which proves that CNN is effective for exploiting local dependency to predict relations.

#### F.1.2 Distance Between Start-to-Start and End-to-End Token Pairs VS. Relational F1

As shown in Figure 3(b), if the distance between the start-to-start and end-to-end relational token pairs is small, the CNN should be helpful. To verify this assumption, we first define the "**Inner Relational Distance**" as follows, for two spans $(s_1, e_1)$ and $(s_2, e_2)$, the relational token pairs are $(s_1, s_2)$ and $(e_1, e_2)$, then the distance between two relational token pairs is calculated as follows

$$d = \max(e_1 - s_1, e_2 - s_2) + 1, \tag{10}$$

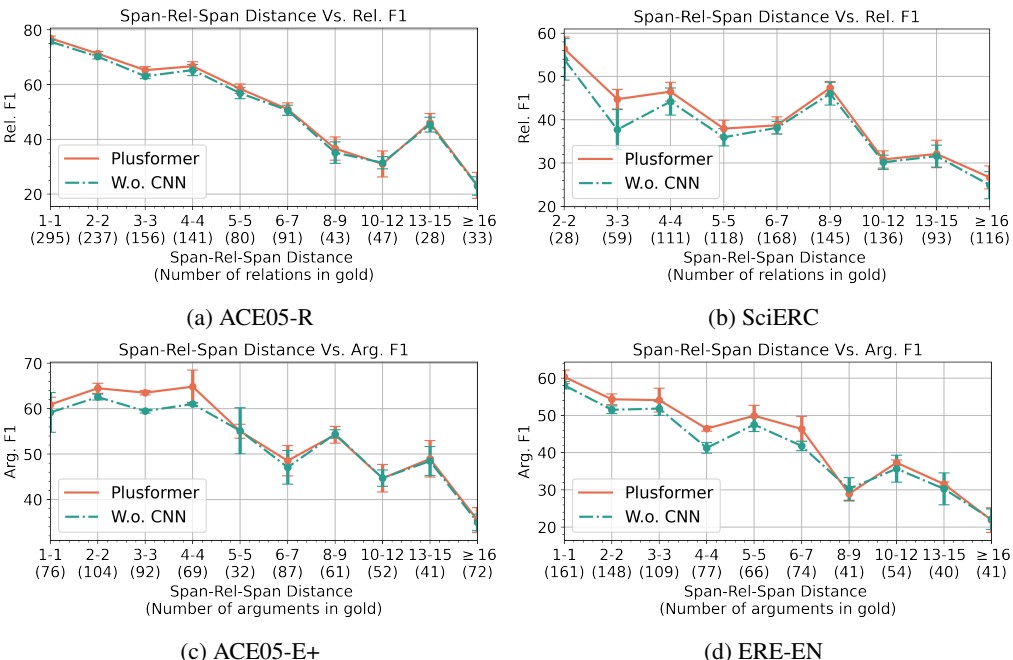

Figure 4: Distance between the relational token pair and its constituent spans (Span-Rel-Span Distance) VS. relational F1 when with or without CNN in Plusformer. The upper and lower figures are for RE and EE tasks, respectively. From the results, it is clear that without CNN, the performance of Plusformer will drop when extracting relations (relation for RE and argument for EE) between nearby spans, while the performance is less effected for relations with further constituent spans. We conjecture this is because the receptive field of CNN is limited to a relatively small distance.

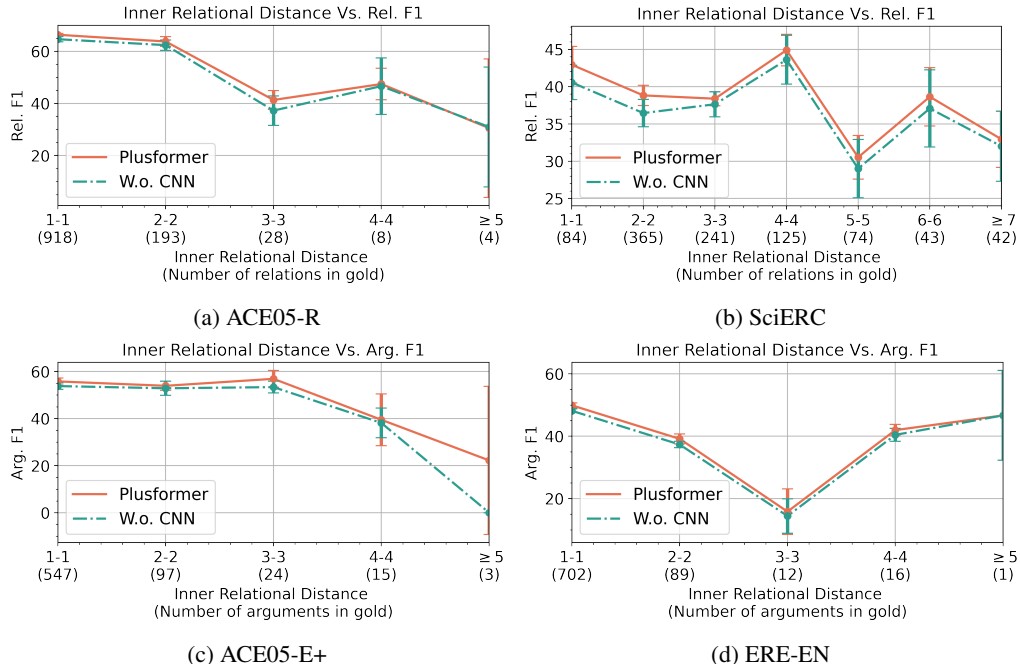

Figure 5: Distance between two relational token pairs of the same span pair (Inner Relational Distance) VS. relational F1 when with or without CNN in Plusformer. The upper and lower figures are for RE and EE tasks, respectively. Since almost all spans are of a length of less than 5, CNN is valuable to model the interaction between start-to-start and end-to-end relational pairs.

where $d$ reveals the distance between start-to-start and end-to-end token pairs, and it is actually decided by the max constituent span length. And its relation with the relational F1 is shown in Figure 5. It is clear that, most of the start-to-start token pair is near to its end-to-end token pair, and CNN takes advantage of this adjacency to make better predictions.

### F.1.3 CNN KERNEL SIZE VS. F1

We study the relation between the kernel size of CNN and F1 performance in Figure 6. We observe that CNN with kernel size 3 obtains the best performance on almost all datasets and tasks. Specifically, reducing CNN kernel size to 1 significantly harms the performance on all datasets, for CNN will lose the capability of interacting with neighboring token pairs. In contrast, F1 also slightly decreases with larger CNN kernel size. We presume that CNN with a larger kernel size may introduce more noise and harm performance. Therefore, we choose kernel size 3 for all datasets.

### F.2 IS CNN ALL WE NEED?

Since CNN is so effective in the Plusformer, it is natural to ask whether it is enough only to use CNN. Therefore, we conduct experiments on models without the plus-shaped self-attention and named this model CNN-IE. We conduct experiments for CNN-IE in six datasets, and results are listed in Table 10 and Table 11. Only with the CNN module can the model achieve SOTA or near SOTA performance in all six datasets, which depicts the effectiveness of the proposed token-pair decomposition and CNN module. However, it still lags behind the UTC-IE model, which reveals the necessity of the PlusAttention.

### F.3 POSITION EMBEDDINGS

The RoPE embedding aims to help token pairs be aware of the spatial relationships between each other, and the triangle position embedding tries to enable spans to be informed of their areas in the feature map. From Table 10 and Table 11, the position embeddings enhance the span extraction

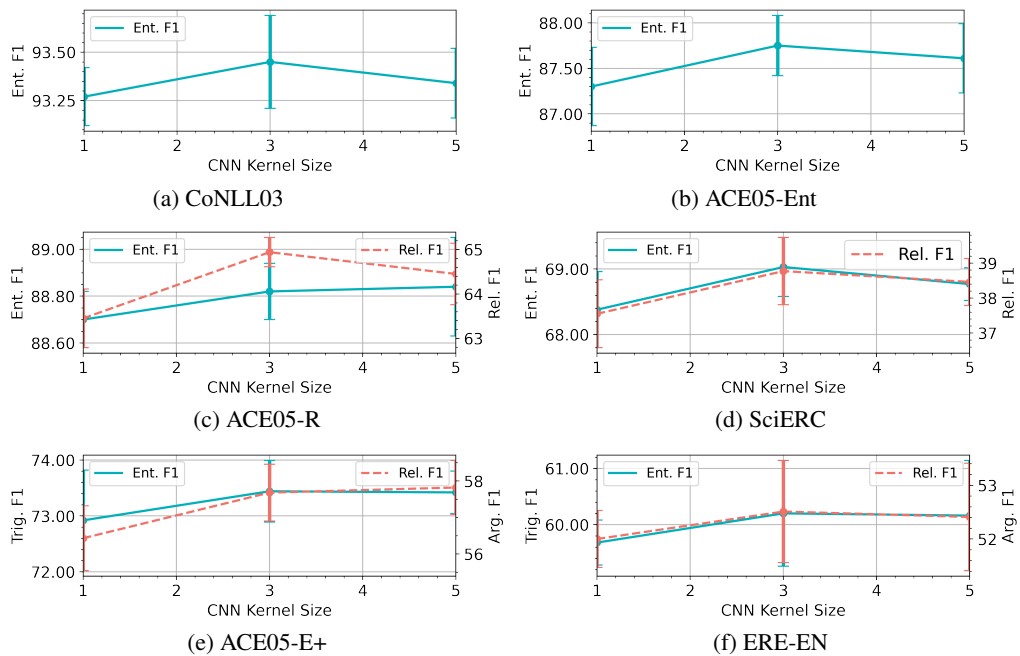

Figure 6: The performance varies with the kernel size of CNN. NER, RE and EE results are listed from top to bottom. CNN with kernel size 3 has the best performance over almost all datasets.

and relational extraction for 0.39, 0.64 averagely. Besides, in Figure 7, we show that the position embeddings can help the model exploit the distance bias to improve the performance of relational extraction.

### F.4 AXIS-AWARE PLUS-SHAPED SELF-ATTENTION

Lastly, we study the effect of PlusAttention. We present an example to delineate why the axis-aware is valuable for span extraction and relational extraction in Figure 8. From Figure 8, axis-aware should be worthwhile no matter what the task is, span extraction or relational extraction. As expected, from Table 10 and Table 11, if we discard the axis-aware in Plusformer, the average performance of span extraction and relational extraction diminish 0.28 and 0.86, respectively, which reveals the necessity of axis-aware in the PlusAttention module.

Besides, we show two case studies of the plus-shaped attention in Figure 9. The sentences are from the test dataset of ACE2005-Ent and ACE2005-R. Both cases put larger attention scores on informative token pairs.

### F.5 NUMBER OF PLUSFORMER VS. F1

We study the relation between the number of Plusformer layers and F1 performance in Figure 10. For the NER datasets, we use two layers of Plusformer, and for the RE and EE we use three.

## G COMPARISON WITH GLAD

Jiang et al. (2020) claims that many NLP tasks can be regarded as the span prediction and prediction of relations between pairs of spans (named as span extraction and relational extraction in our paper), which is conceptually similar to our insights. However, our work is fundamentally distinct from theirs on both formulation and model architecture. Jiang et al. (2020) classify various NLP tasks into two separate tasks and design different modules for them. To contrast, we unify all traditional IE tasks into a single formulation, namely token-pair classification. Therefore, we only need one model for all tasks. Besides, Jiang et al. (2020) simply use the concatenation of the start and end

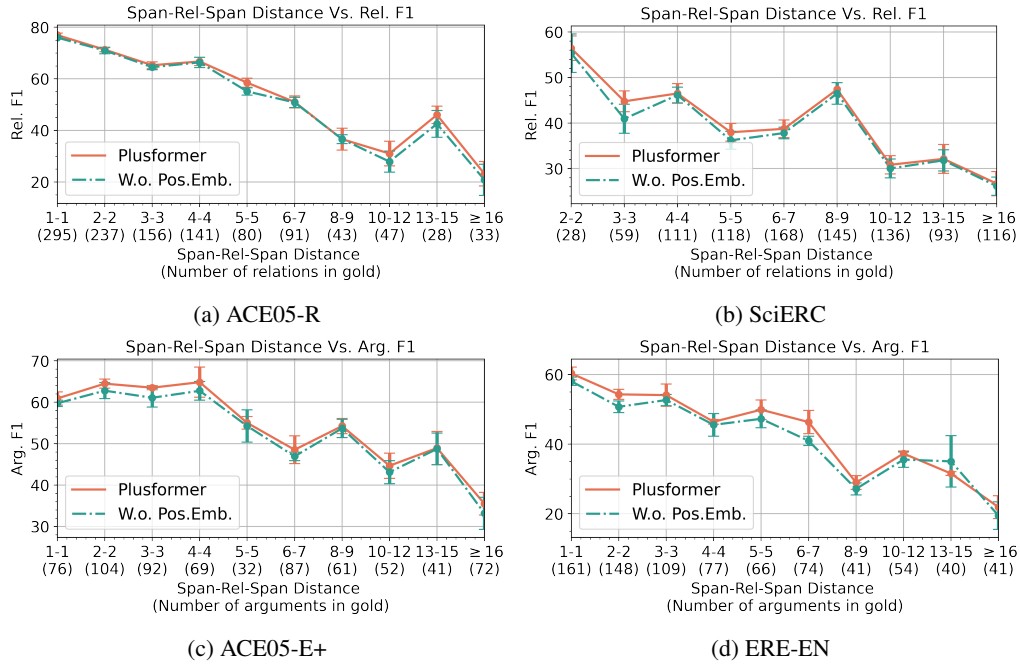

(a) ACE05-R

(b) SciERC

(c) ACE05-E+

(d) ERE-EN

Figure 7: Distance between the relational token pair and its constituent spans (Span-Rel-Span Distance) VS. relational F1 when with or without position embeddings in Plusformer. The upper and lower figures are for RE and EE tasks, respectively. Without position embeddings, the relational performance is lower almost in all "Span-Rel-Span" distances. We presume this is because, with position embedding, Plusformer can exploit the distance inductive bias to determine the relations.

Table 12: Results comparison between a similar work GLAD and UTC-IE on NER, RE, SRL and ABSA. We leverage BERT-base as base model for fair comparison. GLAD performs NER and RE jointly on WLP dataset, and report them separately. We use the same settings as theirs. ♣ means that the UTC-IE without Plusformer surpasses previous SOTA performance.

|  | **NER** WLP | **RE** WLP | **OIE** OIE2016 | **SRL** OntoNotes | **ABSA** SemEval14 |
|---|---|---|---|---|---|
| GLAD (Jiang et al., 2020) | 78.1 | 64.7 | 36.7 | 83.3 | 70.8 |
| **UTC-IE** | $\mathbf{82.51}_{\pm 31}$ | $\mathbf{68.57}_{\pm 54}$ | $\mathbf{37.90}_{77}$ | $\mathbf{84.90}_{\pm 39}$ | $\mathbf{73.53}_{\pm 45}$ |
| -Plusformer | $79.47^{\clubsuit}_{\pm 42}$ | $66.07^{\clubsuit}_{\pm 37}$ | $36.73^{\clubsuit}_{\pm 79}$ | $83.75^{\clubsuit}_{\pm 35}$ | $71.80^{\clubsuit}_{\pm 52}$ |

token representations to represent a span, and for relations, they concatenate the head and tail span representations. Therefore, in their work, the interaction between spans are weak. In our work, we obtain the feature matrix of all token pairs and add well-designed Plusformer module on top of all token pairs, where token pairs can interact with others thoroughly.

In order to prove the superiority of our reformulation and UTC-IE model, we make a fair comparison on several tasks from the GLAD benchmark (Jiang et al., 2020). We choose 3 additional IE tasks, including Open Information Extraction (OIE), Semantic Role Labeling (SRL) and Aspect Based Sentiment Analysis (ABSA), and NER and RE. We use WLP (Hashimoto et al., 2017) on NER and RE, OIE2016 (Stanovsky & Dagan, 2016) on OIE, OntoNotes (Pradhan et al., 2013) on SRL and SemEval14 (Pontiki et al., 2014) on ABSA. The detailed experimental settings are the same as those in GLAD, to ensure a fair comparison. Results are present in Table 12.

The table shows that UTC-IE outperforms GLAD on all chosen tasks exceedingly, with +2.76 improvement on average. Moreover, we observe that UTC-IE without Plusformer also surpasses GLAD benchmarks on all tasks with +0.84 improvement on average, which proves the superiority of our unified reformulation.

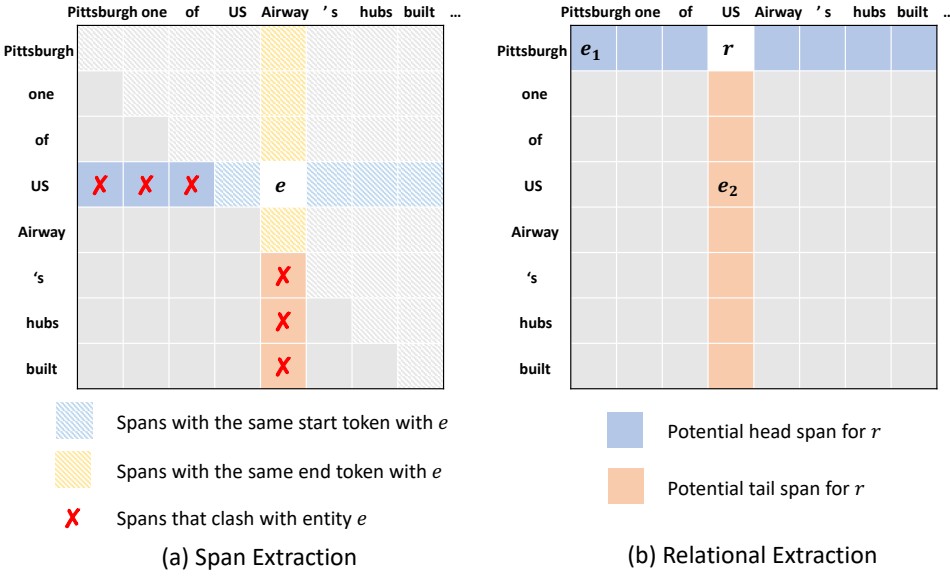

Figure 8: Examples to show why axis-aware is meaningful for IE tasks. In the left figure, spans in $e$'s vertical direction share the same end token as $e$ except for spans in the lower triangle, since they clash with $e$ in the back (because "Airway" is the end token of $e$ but the start token for these spans); spans in $e$'s horizontal direction have common start token as $e$, but not spans in the lower triangle, because they clash with $e$ in the front (since "US" is the start token of $e$ but the end token for these spans). Therefore, both the axis-aware and triangle position embedding are crucial for spans to figure out their relationships with each other. In the right figure, for a relational token pair, the spans from its horizontal direction must be the head span, while the tail span must come vertically. Thusly, axis-aware is informative for relational extractions.

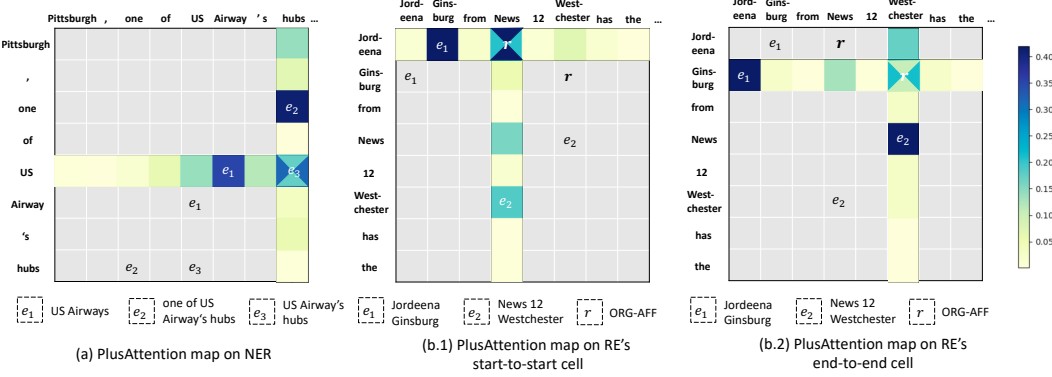

Figure 9: Two case studies of the PlusAttention. The horizontal and vertical attention scores are from the horizontal and vertical self-attentions of last layer of Plusformer. The center cells are with two colors, one for the horizontal attention scores and the other for the vertical attention scores. For NER, the center cell attends more on other entities. And for RE, the center relational cell attends more on its constituent entities.

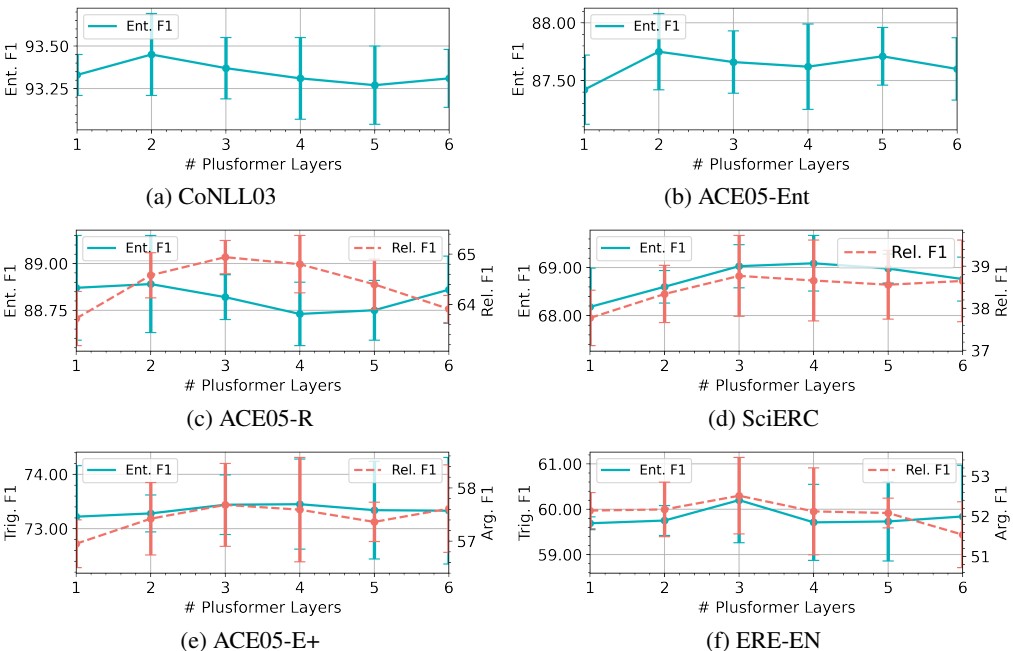

(a) CoNLL03

(b) ACE05-Ent

(c) ACE05-R

(d) SciERC

(e) ACE05-E+

(f) ERE-EN

Figure 10: The performance varies with the number of Plusformer layers. NER, RE and EE results are listed from top to bottom. For the NER tasks, the performance peaks at the two layers of Plusformer, and for RE and EE, the performance plateaus after three layers of Plusformer.

