# OpenReview forum: "UTC-IE: A Unified Token-pair Classification Architecture for Information Extraction"
_ICLR.cc/2023/Conference — Submitted to ICLR 2023_

### Official Review · Reviewer_7KpY · 2022-10-20

**Confidence:** 4
**Clarity, Quality, Novelty And Reproducibility:** See above.
**Correctness:** 3
**Technical Novelty And Significance:** 2
**Empirical Novelty And Significance:** 2
**Recommendation:** 3

**Strength And Weaknesses:**

See above.

**Summary Of The Paper:**

This paper introduces a unified information extraction (UIE) framework based on the token pair classification idea. The UIE solution allows modeling all types of IE tasks such as NER, RE and EE with just one unified model. Also the authors propose a Plusformer on top of the token-pair feature matrix for better feature learning. They show their method advances better efficacy and efficiency. The proposal of UIE simplified with a token pair scheme is technically sound, and also the experiments are solid. However, I'm afraid this paper may suffer from some key flaws, with respect to the topic of UIE, which I list bellow.


1. Overall, this paper made limited contributions.

a. Token pair classification, aka., table filling/grid tagging, is one of the most typical and long-standing task modeling schemes for the information extraction jobs in NLP community.
Also, the paper claims to advance with unifying all IE tasks, yet the idea of token pair classification has been well explored in NER [2], RE [1] and EE [3].
Thus this paper leverages such a method for UIE, which in the perspective of technique, contributes less.

b. The proposed Plusformer is simply the combination of CNN layer with axis-aware addition. The module is designed for modeling the local interaction. However as far as I know, this will fall short on modeling some of the long-range dependences between elements for especially the RE/EE tasks [5].



2. Problematic/unconvincing motivations

a. The authors indicate the issues of the existing works: "However, these models are either complex to be designed or time-consuming to decode", which could be problematic. For example, generative seq2seq methods are very easy and straightforward, while graph-based methods enjoy high running efficiency due to high parallel architecture.

b. The motivation and designing detail of the use of token pair in this paper is essentially the same as the recent work [2], while the authors seem to skip [2] without acknowledging it, which is unacceptable.

c. Suspicious unified IE. I don't feel that the task modeling should be called as unified information extraction, since it seems different task types (NER/RE/EE) come with one specific task formalization. [4] models all different IE tasks with just one real single unified formulation under a well-designed structured extraction language/expression.

And also the paper writing or technique description is not clear, e.g.,
1) Y [s, e, t] = Y [e, s, t] , what does the [*] mean?
2) how to solve the issue of overlapping elements in NER/RE/EE with your formulization?
3) the task formulization lacks necessary examples for certain IE tasks.


3. Another potential problem in the technique method part: one of the key reasons to propose UIE in IE community lies in the capability of knowledge transfer, i.e., pre-training once and transferring everywhere, to naturally support few-shot learning for downstream IE task. But the UIE in this paper seems to fail to well support this advantage; they more likely perform unified IE task modeling, and nothing more. This would limit the real merit of UIE.


4. As an extension of token pair based IE method, this work didn't survey the existing works for this line, e.g., differences.
Also some critical existing works are overlooked [1,2,3].



[1] Modeling Joint Entity and Relation Extraction with Table Representation. EMNLP 2014.

[2] Unified Named Entity Recognition as Word-Word Relation Classification. AAAI 2022.

[3] OneEE: A One-Stage Framework for Fast Overlapping and Nested Event Extraction. COLING 2022.

[4] Unified Structure Generation for Universal Information Extraction. ACL 2022.

[5] Graph Convolution over Pruned Dependency Trees Improves Relation Extraction. EMNLP 2018.


**Summary Of The Review:**

See above.

---

> ### Author Response · Authors · 2022-11-09
> **Response to Review 7KpY (Contributions and Movitations)**
>
> Thanks for your detailed comments and triggering discussions on our reformulation.
>
> ### Limited contributions:
>
> > Q1: The idea of token pair classification has been well explored in NER [2], RE [1] and EE [3].
>
> A1: Firstly, although the three papers are all span-based, the formulations of them are different. For example, NER[2] decomposes entity extraction as a chain of token-pair classification, RE[1] first formulates span extraction as sequence tagging and then classifies relations between spans, and EE[3] is actually our contemporary work. All of these formulations are task-specific. Different from them, we propose a simple way to unify all IE tasks into token-pair classification.
>
> Secondly, most of the previous methods only consider shallow interactions between spans in 1D sequential dimension. However, based on reformulation, we design Plusformer to exploit deep interactions between token pairs in 2D feature matrix, which has been shown to be effective by comprehensive experiments and ablation studies.
>
> > Q2: Plusformer is designed for modeling the local interaction. However, as far as I know, this will fall short on modeling some of the long-range dependencies between elements for especially the RE/EE tasks [5]
>
> A2: Plusformer can model both local interaction by CNN and global interaction by PlusAttention. We think you may doubt how can Plusformer deal with relations whose head and tail entities that are far from each other. Firstly, in the feature matrix, one relation is decided by two cells (start-to-start and end-to-end token pairs). For each cell, the central relational token pair’s two constituent spans locate in the plus-shaped orientation, and can interact with others by PlusAttention, as shown in Figure 8 and Figure 9.
>
> Besides, as proved in Equation (10), the distance between two relational cells is only decided by the maximum of constituent entities' length, which is mostly less than 4 in IE tasks (shown in Figure 5). Therefore, two relational token pairs can interact with each other thoroughly by CNN.
>
> ### Problematic/unconvincing motivations:
>
> > Q1: "However, these models are either complex to be designed or time-consuming to decode", which could be problematic.
>
> A1: We are sorry for the confusion for our lack of clarity. We want to claim that generative models are time-consuming to decode for they are auto-regressive, while graph-based models need elaborate and complex design. We have changed our expression in Introduction.
>
> > Q2: The motivation and designing detail of the use of token pair in this paper is essentially the same as the recent work[2].
>
> A2: There are giant differences between [2] and our work.
>
> For the formulation side, [2] only focuses on unified NER tasks by a chain of neighboring word-to-word pairs, which do not consider the relation between spans. However, UTC-IE use the start-to-end token pair to represent a span, and the start-to-start and end-to-end pair to represent a relation.
>
> For the model side, we propose Plusformer to fully exploit the interaction between token pairs after the reformulation, and experiments in five NER datasets show that UTC-IE surpasses [2] by 0.57 averagely.
>
> > Q3: I don't feel that the task modeling should be called as unified information extraction, since it seems different task types (NER/RE/EE) come with one specific task formalization.
>
> A3: We think unified tasks refer that we only need to design one single module (without designing any task-specific modules) to solve all tasks. To achieve this goal, we first need to formulate all tasks under the same input and output formats. UIE[4] and TANL[6] unify all tasks in a Seq2Seq way, and convert all tasks' input and output into sequences, while this does not mean for each task, the output is the same. For example, in UIE, when extracting spans, the output is one level of bracket and when extracting relations, the output is two levels of brackets (As depicted in Figure 2(b) of [4]).
>
> Based on this standard, UTC-IE indeed uses one module to solve all IE tasks (without any task specific module) after reformulating all IE tasks into a uniform token-pair classification way.

---

> > ### Author Response · Authors · 2022-11-09
> > **Response to Reviewer 7KpY (paper writing or technique description)**
> >
> > ### Paper writing or technique description:
> >
> > > Q1: Y[s, e, t] = Y[e, s, t] , what does the [*] mean?
> >
> > A1: [\*] means matrix indexing. Specifically, $Y[s, e, t]$ means the score of directional $(s,e)$ token pairs with type $t$.
> >
> > > Q2: how to solve the issue of overlapping elements in NER/RE/EE with your formulization?
> >
> > A2: We cannot confirm what overlapping elements refer to. We have the following two comprehensions.
> >
> > First, as for the overlapping between spans in single task, they are decomposed into different cells in feature matrix and will not cause conflict in reformulation. Besides, the detailed methods of resolving conflicts during decoding are presented in Appendix B.
> >
> > Second, as for the overlapping between spans cross tasks, we decode different task separately and will not face the overlapping issue in single IE. In joint IE, as described in Appendix B, all entity spans are entity candidates of RE and argument candidates of EE, thus spans will not overlap with others.
> >
> > > Q3: one of the key reasons to propose UIE in IE community lies in the capability of knowledge transfer, i.e., pre-training once and transferring everywhere, to naturally support few-shot learning for downstream IE task. But the UIE in this paper seems to fail to well support this advantage; they more likely perform unified IE task modeling, and nothing more.
> >
> > A3: We think that modeling IE task uniformly and effectively is essentially meaningful. After unifying, we do not need to design task-specific models for different IE tasks, but can solve all IE tasks with a single model.
> > Besides, we also experiment on joint IE, which solves three IE tasks simultaneously with our model, and achieve better performance than that on single IE in Table 2. It convincingly proves that unifying different tasks under the same model has the capability of knowledge sharing between various tasks.
> > Finally, as described in Conclusion, the token-pair reformulation opens a new angle to broaden the communication to the 2-D feature matrix, and future work can exploit the few-shot scenario in a similar way.
> >
> > > Q4: As an extension of token pair based IE method, this work didn't survey the existing works for this line, e.g., differences. Also some critical existing works are overlooked [1,2,3].
> >
> > A4: The differences between the mentioned work has detailed in "A1" of "Limited contributions".
> >
> >
> > [1] Modeling Joint Entity and Relation Extraction with Table Representation. EMNLP 2014.
> >
> > [2] Unified Named Entity Recognition as Word-Word Relation Classification. AAAI 2022.
> >
> > [3] OneEE: A One-Stage Framework for Fast Overlapping and Nested Event Extraction. COLING 2022.
> >
> > [4] Unified Structure Generation for Universal Information Extraction. ACL 2022.
> >
> > [5] Graph Convolution over Pruned Dependency Trees Improves Relation Extraction. EMNLP 2018.
> >
> > [6] STRUCTURED PREDICTION AS TRANSLATION BETWEEN AUGMENTED NATURAL LANGUAGES. ICLR 2021.

---

### Official Review · Reviewer_j1ou · 2022-10-25

**Confidence:** 4
**Correctness:** 3
**Technical Novelty And Significance:** 2
**Empirical Novelty And Significance:** 2
**Recommendation:** 5

**Clarity, Quality, Novelty And Reproducibility:**

 - In the abstract, it is a stretch to claim to unify all IE tasks as there are only 4 tasks in the experiment setting.

 - Using the start and end token of the span as a representation is common practice for information extraction and relation extraction.

 - It is not clear why the CNN is limited to kernel size 3.

 - In Table 4, it is not clear what is the difference between UTC-IE without “axis-aware” and CNN-IE which does not have PlusAttention.


**Strength And Weaknesses:**

Strengths
 - The speed comparison in section 4.4 empirically supports the computational efficiency of the proposed model.

Weaknesses
 - The motivation for the token-pair reformulation is not convincing as there is no fundamental challenge in unified IE that is being addressed.
 - There is limited novelty as the model combines several existing components (position embeddings, CNN) that are not specific to the token-pair formulation.
 - The setting of unified IE is limited as previous works (e.g. GLAD benchmark) included more tasks such as coreference, parsing and aspect-based sentiment analysis.
 - The SpanRel model (Jiang et al., 2020) is similar to this work in motivation and formulation, but the authors’ comparison is superficial and SpanRel is missing from the experiment results.


**Summary Of The Paper:**

To address the information extraction (IE) setting for multiple tasks, the authors propose to formulate the IE problem as token-pair classification. The proposed formulation represents the spans and span-pair relations through token-pair interactions which is similar to table-filling approaches. The experiment results show benefits compared to prior methods on 10 datasets.

**Summary Of The Review:**

Although the proposed approach is relatively simple and can be applied to multiple IE tasks, there are issues with the motivation and novelty of the work.

---

> ### Author Response · Authors · 2022-11-09
> **Response to Reviewer j1ou (Weaknesses)**
>
> Thanks for your thoughtful review and useful suggestions to make our paper better.
>
> ### Weaknesses:
>
> > Q1: The motivation for the token-pair reformulation is not convincing as there is no fundamental challenge in unified IE that is being addressed.
>
> A1: Previous task-specific span-based methods do not consider the interaction between spans seriously, our token-pair reformulation can fundamentally make it easy to enhance interactions between token pairs. As shown in our comprehensive experiments in Table 1 and Table 2, using Plusformer under the token-pair reformulation outperforms SOTA on various IE tasks, with averagely +0.35 improvement on span extraction and +1.26 improvement on relational extraction. We think the significant improvement on performance is fundamentally essential to application tasks such as IE.
>
> > Q2: There is limited novelty as the model combines several existing components (position embeddings, CNN) that are not specific to the token-pair formulation.
>
> A2: We think that whether using existing components cannot define the novelty of a model. Besides, the key point of our paper is to reformulate IE into token-pair classification and consider the interaction between token pairs, instead of proposing bells and whistles modules.
>
> After reformulating into the feature matrix, every module in Plusformer, including the plus-shaped attention, special position embeddings and CNN is well-motivated (numerous detailed analysis can be found in Appendix F). And extensive experiments prove that through these well-established components, we can achieve substantial performance enhancement.
>
> > Q3: The setting of unified IE is limited as previous works (e.g. GLAD benchmark) included more tasks such as coreference, parsing and aspect-based sentiment analysis.
>
> A3: We initially follow IE settings in OneIE[1], which only considers traditional IE tasks. Take your suggestions into consideration, we introduce broader IE tasks in our paper. Following GLAD[2], we add 3 additional tasks, namely open information extraction (OIE), semantic role labeling (SRL) and aspect-based sentiment analysis (ABSA). Besides, we also experiment on WLP dataset for NER and RE. The results are shown below and detailed analysis is shown in Appendix G. UTC-IE outperforms GLAD on all additional datasets significantly with the same pre-trained model.
>
> | Task        | NER       | RE        | OIE       | SRL       | ABSA      |
> | ----------- | --------- | --------- | --------- | --------- | --------- |
> | Dataset     | WLP       | WLP       | OIE2016   | OntoNotes | SemEval14 |
> | GLAD [2]    | 78.1      | 64.7      | 36.7      | 83.3      | 70.8      |
> | UTC-IE      | **82.51** | **68.57** | **37.90** | **84.90** | **73.53** |
> | -Plusformer | 79.47     | 66.07     | 36.73     | 83.75     | 71.80     |
>
> > Q4: The SpanRel model (Jiang et al., 2020) is similar to this work in motivation and formulation, but the authors’ comparison is superficial and SpanRel is missing from the experiment results.
>
> A4: Thanks for your advice. We add the detailed comparison on both formulation and results in Appendix G, and cite SpanRel[2] in Introduction.
>
> For short, SpanRel focuses on classifying various NLP tasks into span extraction and relational extraction, with different formulations and two-stage modules. Specifically, one classification module is taking a span representation and the other classification module is taking relations between two span representations. However, we further unify all IE tasks into one formulation, namely token-pair classification, thus we only need one model for all kinds of IE tasks.

---

> > ### Author Response · Authors · 2022-11-09
> > **Response to Review j1ou (Clarity)**
> >
> > ### Clarity:
> >
> > > Q1: In the abstract, it is a stretch to claim to unify all IE tasks as there are only 4 tasks in the experiment setting.
> >
> > A1: We add more tasks in Appendix G as described in "A3" in "Weaknesses".
> >
> > > Q2: Using the start and end token of the span as a representation is common practice for information extraction and relation extraction.
> >
> > A2: As far as we know, none of them have used this simple formulation to unify all the three IE tasks. We guess one reason might be the weak performance, most of them model spans and relations between spans separately. However, after converting span extraction and relational extraction into token-pair classification uniformly, we can model both tasks under the same feature space. Through the 2D interaction between token pairs, we can unify three IE tasks and achieve SOTA performance.
> >
> > > Q3: It is not clear why the CNN is limited to kernel size 3.
> >
> > A3: Thanks for your advice. We supplement experiments and analysis in Appendix F.1.3 to show the influence of CNN kernel sizes on performance. We conclude that setting kernel size as 3 obtains the best performance on almost all tasks.
> >
> > > Q4: In Table 4, it is not clear what is the difference between UTC-IE without “axis-aware” and CNN-IE which does not have PlusAttention.
> >
> > A4: As described in footnote 5, UTC-IE without "axis-aware" means do not distinguish horizontal and vertical features in PlusAttention. The only change is the combination way of $Z_h$ and $Z_v$ in Equation (4). As for CNN-IE, as described in Section 4.5, we remove the whole PlusAttention module, where CNN is directly on top of the biaffine module.
> >
> >
> > [1] A joint neural model for information extraction with global features. ACL 2020.
> >
> > [2] Generalizing natural language analysis through span-relation representations. ACL 2020.

---

### Official Review · Reviewer_TuyE · 2022-10-25

**Confidence:** 4
**Correctness:** 3
**Technical Novelty And Significance:** 3
**Empirical Novelty And Significance:** Not applicable
**Recommendation:** 8

**Clarity, Quality, Novelty And Reproducibility:**

This work is of good quality and is overall clear and easy to follow. Moreover, the paper proposes an interesting idea to unify all IE tasks into token-pair classification tasks and proposes a relatively new attention mechanism for token-pair feature matrix; therefore, the originality of the work is good.

**Strength And Weaknesses:**

Strengths:
* The paper proposes an interesting idea to unify all IE tasks into token-pair classification tasks.
* The paper has performed extensive experiments on 12 IE tasks and showed the superiority of the proposed method on both performance and decoding speed.
* The paper is overall well written.

Weaknesses:
* The paper proposed a new attention mechanism for token-pair feature matrix. It would be better if the authors could provide some visualization examples on axis-aware attention and location interaction attention. The authors could also provide qualitative examples of how this new attention mechanism helps with performance.
* In equation (4), it is unclear the concatenation of Z_h and Z_v is along which axis.
* In Section 4.3, the author mentioned that “the performance of UTC-IE_single is from trigger/argument F1 of EE”; however, the trig. (and arg. F1) score for UTC-IE_single on ERE-EN in Table 2 is 57.01 (and 48.29), while the trig. (and arg. F1) score for UTC-IE on ERE-EN in Table 1 is 60.20 (and 52.51). According to the appendix, it seems like the difference is due to the usage of different pre-trained models? If so, the authors might want to consider pointing out this in the main content.

**Summary Of The Paper:**

One existing problem in information extraction is that it consists of a wide range of tasks. Previous attempts to unify all IE tasks with one architecture are either complex or time-consuming to decode. To this end, this paper proposes a simple yet effective paradigm for unified IE by interpreting all IE tasks as span extraction and span relation extraction. Specifically, the paper introduces Plusformer, which models the axis-aware interaction and location interaction on top of the token-pair feature matrix. Experiments on 12 IE tasks show that the proposed approach achieves SOTA performance. The proposed model is also much faster than prior unified IE models

**Summary Of The Review:**

The paper proposed an interesting idea to unify all IE tasks into token-pair classification tasks and proposes a relatively new attention mechanism for token-pair feature matrix. Extensive experiments and ablation studies have been performed to demonstrate the superiority of the proposed method on performance and decoding speed. Overall, I believe this paper should be accepted.

---

> ### Author Response · Authors · 2022-11-09
> **Response to Reviewer TyuE**
>
> We greatly appreciate your valuable and positive feedbacks which encourage us considerably.
>
> ### Weaknesses:
>
> > Q1: The authors could also provide qualitative examples of how this new attention mechanism helps with performance.
>
> A1: Thanks for your advice. We add case studies of PlusAttention in Appendix F.4, which depict the horizontal and vertical attention map of NER and RE. Figure 9 shows that the center cell attends more on other entities in NER and attends more on constituent entities in RE, which shows the effectiveness of our proposed plus-shaped attention.
>
> > Q2: In equation (4), it is unclear the concatenation of Z_h and Z_v is along which axis.
>
> A2: We concatenate $Z_h\in R^{L\times L\times c}$ and $Z_v\in R^{L\times L\times c}$ on the last dimension, and results in a $L\times L\times2c$ tensor.
>
> > Q3: In Section 4.3, the author mentioned that “the performance of UTC-IE_single is from trigger/argument F1 of EE”; however, the trig. (and arg. F1) score for UTC-IE_single on ERE-EN in Table 2 is 57.01 (and 48.29), while the trig. (and arg. F1) score for UTC-IE on ERE-EN in Table 1 is 60.20 (and 52.51).
>
> A3: It is caused by two aspects. First, pre-trained models are different, as listed in Table 6. Second, the data processing is different. We follow the data pre-processing code of OneIE[1], and find that data pre-processing has some detailed differences between single IE and joint IE scenario, for example, sentences will be truncated if they are longer than 128 in joint IE scenario but will not in single IE.
>
> [1] A joint neural model for information extraction with global features. ACL 2020.

---

### Author Response · Authors · 2022-11-15
**General Response**

We sincerely appreciate the thoughtful comments by the three reviews!

After discussing with reviewers, we highlight that for tasks, such IE tasks, which are applied widely in various companies, concrete performance gain is of vital importance, and our proposed model unveils new SOTA on 10 datasets across 13 settings. Specially, for span extraction and relational extraction, our methods enhance the SOTA performance by 0.35 and 1.26, respectively. Therefore, this work is meaningful for the IE field.

Besides, we conduct comprehensive analysis (most of them locates in the Appendix because of page limit) to dissect the reasons why our proposed methods achieve good performance. Results show that interactions between token pairs are beneficial, which are almost ignored by previous work. We presume that the absence of interactions in previous work mainly originated from lack of proper formulation and modules to exploit the interaction. In our paper, we reformulate different IE tasks into token-pair classification tasks. Based on the reformulation, we further propose Plusformer which can effectively take advantage of the inductive bias between token pairs.

All in all, we want to stress that exploiting spatial interactions between token-pairs (or spans) is a promising way to advance IE tasks further, our work is a preliminary study to show the effectiveness of this direction. Expanding the exploitation into the 2D feature map can help us take more inspiration from the Computer Vision field.

Moreover, we follow the reviews' useful suggestions and updated the paper, and we summarize the major changes as follows:

1. Add more experiments.
   1. We add more experiments on 4 datasets from GLAD benchmark[1] across 5 IE tasks, such as open information extraction (OIE), semantic role labeling (SRL) and aspect-based sentiment analysis (ABSA) etc, and results significantly surpass those in [1] on all these datasets. (Appendix G)
   2. We add discussions on the influence of CNN kernel sizes on F1, and present quantitive results.  (Appendix F.1.3)
   3. We add two case studies of PlusAttention for NER and RE. (Figure 9)
2. Improve clarity.
   1. We add citation on [1] in Introduction (Section 1) and describe the detailed differences between them in Appendix G.
   2. We modify the description of previous unified IE models in Introduction for clarity. (Section 1)

[1] Generalizing natural language analysis through span-relation representations. ACL 2020.

---

### Author Response · Authors · 2022-11-15
**Looking Forward to Your Reply!**

Dear Reviewers,

Thanks again for your thoughtful feedbacks and comments! Since the discussion period is ending soon, we would sincerely appreciate it if you could let us know if you are satisfied with our responses. We will be glad to address any remaining concerns.

Sincerely,

Paper 3809 Authors

---

### Decision · Program_Chairs · 2023-01-20

**Decision:**

Reject

**Justification For Why Not Higher Score:**

While accepting this paper would showcase a previously non-existing neural architecture, this would have relatively small novelty and it won't be clear how different elements of the architecture contribute to the performance over the IE task.

**Justification For Why Not Lower Score:**

N/A

**Metareview: Summary, Strengths And Weaknesses:**

This paper presents a transformer-based architecture for jointly performing three information extraction (IE) tasks: named entity recognition (NER), relation extraction (RE), and event extraction (EE). The  main contribution of this work is the Plusformer, an attention mechanism that is well-suited for capturing pairwise relationships over the Biaffine model.

Strengths:
- A new architecture for IE tasks.
- An extensive evaluation over a large set of benchmarks showing improvements over baselines.

Weaknesses:
- The various design decisions for Plusformer are not well ablated or motivated. For example, is the 3x3 CNN a good choice? What if a traditional transformer output was used instead? Why 3x3 and not 4x4 for example?
- Similarly, is the Biaffine model necessary after a PLM and before a Plusformer? Transformers are good at capturing (all) pairwise relationships among inputs, could the Plusformer be combined just with the standard PLM?
- What is the effect of different pretrained LMs (PLMs) to the performance of the model? What is the effect of the size of the PLM to the target tasks?

Overall, I recommend that this paper is not accepted at its current state as some additional evaluations and ablations would be needed to illustrate the advantages of the proposed model.

Minor notes:
- The Plusformer attention has a resemblance to many of the sparse transformer architectures (e.g. BigBird, Axial Transformer see Tay, Yi, et al. "Efficient transformers: A survey." ACM Computing Surveys (CSUR) (2020).). This should be acknowledged in Sec 3.2.

**Summary Of Ac-Reviewer Meeting:**

Given that the AC of the paper was reassigned two days before the deadline no meeting could be scheduled.